# GUMBEL DISTILLATION FOR PARALLEL TEXT GENERATION

**Chi Zhang**[*]  **Xixi Hu**[*]  **Bo Liu**   **Qiang Liu**
Department of Computer Science
The University of Texas at Austin
`{chizhang, hxixi, bliu, lqiang}@cs.utexas.edu`

## ABSTRACT

The slow, sequential nature of autoregressive (AR) language models has driven the adoption of parallel decoding methods. However, these non-AR models often sacrifice generation quality as they struggle to model the complex joint distribution of token sequences. To narrow this performance gap, we introduce Gumbel Distillation, a novel distillation technique that enables parallel decoders to learn this distribution effectively. Our method leverages the Gumbel-Max trick to create a deterministic mapping from a latent Gumbel noise space to the output tokens of a high-performing AR teacher. As a model-agnostic technique, Gumbel Distillation seamlessly integrates with diverse parallel decoding architectures, including MDLM and BD3-LM. Experiments on LM1B and OpenWebText show that Gumbel Distillation substantially improves the generation quality of parallel language models, achieving a 30.0% improvement in MAUVE Score and 10.5% in generative perplexity over MDLM trained on OpenWebText dataset.

## 1 INTRODUCTION

Autoregressive (AR) language models have set the standard for text generation (Radford et al., 2019; Brown et al., 2020; Touvron et al., 2023), but their sequential, token-by-token inference process introduces significant latency, hindering their use in real-world applications. To address this bottleneck, various parallel decoding methods have emerged, including Masked Diffusion Language Models (MDLMs) (Shi et al., 2024; Sahoo et al., 2024; Arriola et al., 2025; Ou et al., 2025; Nie et al., 2025) and Multi-Token Prediction (MTP) (Cai et al., 2024; Gloeckle et al., 2024; Liu et al., 2024a). These non-autoregressive approaches accelerate inference by generating multiple tokens simultaneously. However, this speedup often comes at a cost of degradation in generation quality.

This performance gap originates from a fundamental challenge in non-AR modeling (Gu et al., 2017): the difficulty of learning the joint probability distribution of an entire sequence. AR models elegantly handle this by factorizing the distribution into a product of conditional probabilities using the chain rule (Vaswani et al., 2017), capturing the dependency of each token on its predecessors. In contrast, to enable simultaneous prediction, parallel decoders operate under a naive assumption of conditional independence among the tokens generated in a single step. Previous works (Xiao et al., 2022; Liu et al., 2024b; Xu et al., 2024; Song & Zhou, 2025; Wu et al., 2025; Kim et al., 2025) have pointed out that the simplification, while necessary for parallelism, disrupts the natural sequential dependencies of language. For example, predicting the phrase "San Francisco" requires knowing that "Francisco" is highly dependent on the co-occurrence of "San" in the same block. When failing to capture these intricate local dependencies, parallel models can produce errors such as token repetition, resulting in text that is less coherent or grammatically flawed.

In this work, we aim to narrow this quality gap by fundamentally improving the capacity of parallel decoders without sacrificing the speed of parallel decoding. We argue that the key issue is how to make the learning problem for the non-AR student model easier. Instead of asking the model to learn the complex distribution of language from scratch, could we provide a *"blueprint"* from a powerful AR teacher that hints how a specific output sequence was formed?

To that end, we introduce *Gumbel Distillation*, a novel distillation framework that enables a non-AR student to learn the complex token dependencies captured by an AR teacher. The core of our method is the Gumbel-Max trick, which we use to establish a deterministic mapping from a simple

---

[*]Equal contribution.

Gumbel noise distribution to the target output sequence from the AR teacher. The resulting noise vector serves as the latent "blueprint" to its paired token sequence, uniquely encoding the sampling decisions from the teacher. By training the student to reconstruct the text conditioned on the Gumbel noise, we turn the difficult joint-distribution matching problem into a supervised learning problem.

A key advantage of our approach is its simplicity and general applicability. Because the distillation process is framed as adding a conditional input, Gumbel Distillation can be seamlessly integrated as a plug-and-play module into a wide variety of existing parallel decoding architectures. It does not require complex changes to the model's structure, making it a flexible tool for enhancing different frameworks. We demonstrate this by applying Gumbel Distillation to state-of-the-art architectures, including MDLM (Sahoo et al., 2024), BD3-LM (Arriola et al., 2025) and Medusa (Cai et al., 2024), showing it serves as a simple yet powerful complement. Our experiments validate that Gumbel Distillation significantly improves generation quality, enabling high-quality multi-token decoding.

Our main contributions are as follows:

- We propose Gumbel Distillation, a novel framework for distilling knowledge from an AR teacher into a parallel student. The method enables the student to model the distribution more accurately by learning a direct mapping from Gumbel noise to the teacher's output and therefore better approximating the joint token dependencies.
- To demonstrate the simplicity and effectiveness of our approach, we integrate it with state-of-the-art parallel decoding frameworks, including MDLM, BD3-LM and Medusa, improving their performance while making minimal architectural changes.
- Our extensive experiments validate that our method successfully alleviates the gap between autoregressive quality and parallel decoding efficiency, achieving state-of-the-art results in high-quality, accelerated text generation with parallel decoding.

## 2 RELATED WORK

**Diffusion Language Models** Inspired by the success of diffusion models in visual domains (Ho et al., 2020; Song et al., 2021; Liu et al., 2022; Lipman et al., 2022), a number of works have explored diffusion language models (DLMs) by extending the technique to text domains. An intuitive approach is to model text in a continuous space and apply diffusion directly (Li et al., 2022; Gong et al., 2023; Han et al., 2022; Dieleman et al., 2022). To fit the discrete nature of text, another approach focuses on using discrete processes featuring forward and reverse dynamics, with the foundational work laid out by D3PM (Austin et al., 2021). Numerous variants follow (Hoogeboom et al., 2021; He et al., 2022; Campbell et al., 2022; Sun et al., 2022; Chen et al., 2022; Wu et al., 2023; Zheng et al., 2023; Gat et al., 2024), with masked diffusion language models being a specific focus for recent research (Chen et al., 2024; Shi et al., 2024; Nie et al., 2024; Zheng et al., 2024; Lou et al., 2024; Ou et al., 2025; von Rütte et al., 2025; Kim et al., 2025) due to their strong performance. Notably, MDLM (Sahoo et al., 2024) derived a simplified objective for masked diffusion that connects the operation of unmasking tokens to a true generative process. BD3-LM (Arriola et al., 2025) builds upon MDLM to introduce a hybrid approach that models sequences in a blockwise manner while applying diffusion within each block, enabling flexible length generation and improved inference efficiency through kv-caching. In our paper, we apply Gumbel Distillation to these two architectures as examples to show the effectiveness and flexibility of our method.

Encouraged by the above works, recent efforts have scaled diffusion language models to billions of parameters (Nie et al., 2025; Gong et al., 2025; Ye et al., 2025). Furthermore, commercial success like Seed Diffusion (Song et al., 2025), Mercury Coder (Labs et al., 2025) and Gemini Diffusion (DeepMind, 2025) have also demonstrated their practical viability, achieving performance comparable to state-of-the-art AR models with significantly faster speed of inference.

**Multi-Token-Prediction** Multi-token-prediction (MTP) strategies (Cai et al., 2024; Gloeckle et al., 2024; Liu et al., 2024a; Samragh et al., 2025) enable faster, flexible sampling of AR language models. However, current MTP models often predict the probability of each token in a block independently, which is an issue also relevant to diffusion language models. Song & Zhou (2025); Zhu et al. (2025) argues that this naive conditional independence assumption greatly limits the capacity of the model distribution. Although the issue can be alleviated via corrections to the inference procedure (Gu et al., 2017; Leviathan et al., 2023; Santilli et al., 2023; Shi et al., 2024) such as speculative decoding, and most notably recent works like adaptive parallel decoding (Israel et al., 2025),

our Gumbel Distillation method points to a fundamental solution that in principle allows the model to train and infer from the true joint distribution.

**Knowledge Distillation for Parallel Decoding**   A key application of knowledge distillation is to accelerate inference by training a parallel student model to mimic a powerful AR teacher. This line of research was pioneered by Gu et al. (2017), which introduced sequence-level knowledge distillation to train a parallel student on the outputs of an AR teacher. This technique spurred the development of parallel models for machine translation (Ghazvininejad et al., 2019; Gu et al., 2019). More recently, some work (Gu et al., 2023; Kou et al., 2024) have adapted the idea for general language models. Different from these works, our proposed Gumbel Distillation distills from the AR teacher's internal sampling process, which allows the parallel student to learn the full probability landscape.

## 3  BACKGROUND

**The Challenge of Parallel Decoding**   An AR language model, denoted as $p^*$, generates a sequence of tokens $\boldsymbol{x}^{1:n} = (x^1, \cdots, x^n)$ one at a time. [1] The probability of the sequence is factorized as $p^*(\boldsymbol{x}^{1:n}) = \prod_{i=1}^{n} p^*(x^i|\boldsymbol{x}^{<i})$, where $\boldsymbol{x}^{<i}$ represents the preceding tokens. This sequential factorization effectively captures the dependencies between tokens, leading to high-quality output, but at the cost of slow, iterative inference.

In contrast, *parallel decoding* methods accelerate this process by generating multiple tokens at once. Instead of a single token $x^i$, these models predict a *subset* of all tokens, denoted by $\boldsymbol{x}^{\mathcal{I}}$, where $\mathcal{I}$ is a set of token indices. The already generated tokens $\boldsymbol{x}^{\neg\mathcal{I}}$, serve as the context. This paradigm applies to various parallel decoding architectures; for instance, in MTP (Cai et al., 2024; Gloeckle et al., 2024; Liu et al., 2024a), $\mathcal{I}$ is the set of indices for a block of future tokens, and in MDLMs (Shi et al., 2024; Sahoo et al., 2024; Arriola et al., 2025; Ou et al., 2025; Nie et al., 2025), $\mathcal{I}$ is the set of indices for the tokens to unmask. However, to enable simultaneous prediction for all indices in $\mathcal{I}$, these models are often forced to adopt a conditional independence assumption. They approximate the true, sequentially dependent distribution of the AR model $p^*$ with a block-wise factorized one:

$$p_\theta(\boldsymbol{x}^{\mathcal{I}}|\boldsymbol{x}^{\neg\mathcal{I}}) = \prod_{i\in\mathcal{I}} p_\theta(x^i|\boldsymbol{x}^{\neg\mathcal{I}}).$$

This assumption ignores the dependencies within the target set $\boldsymbol{x}^{\mathcal{I}}$, and as a result, these architectures fail to capture the local structure essential for natural language. We aim to mitigate the quality degradation caused by this factorization.

**Gumbel-Max Trick**   The Gumbel-Max trick (Gumbel, 1954) is a reparameterization technique for sampling from a categorical distribution. To sample from a category $X \in \{1, \ldots, V\}$ defined by the Boltzmann distribution with a vector of logits $\boldsymbol{l} = (l_1, \cdots, l_V) \in \mathbb{R}^V$:

$$P(X = k) = \exp(l_k) \Big/ \sum_{j=1}^{V} \exp(l_j),$$

we can first draw i.i.d. standard Gumbel noise $\{\xi_k \sim \mathcal{G}(0,1)\}_{k=1}^{V}$, and then compute:

$$Y = \arg\max_{k} (l_k + \xi_k).$$

The resulting sample $Y$ has the exact *same* distribution as $X$. In practice, a common method to sample $\xi \sim \mathcal{G}(0,1)$ is via inverse transform sampling, where we first sample a Uniform variable $u \sim \mathcal{U}[0,1]$, and then calculate the Gumbel noise as $\xi = -\log(-\log u)$.

One crucial insight here is that we can reframe the sampling process: the randomness is externalized into the Gumbel noise vector $\boldsymbol{\xi} = (\xi_1, \ldots, \xi_V) \in R^V$. While categorical sampling is stochastic, the argmax function in Gumbel-Max is fully *deterministic* once the logits and the noise are known.

## 4  METHOD

Our primary objective is to design a parallel decoder, $p_\theta$, such that its output can match the high-quality output of a powerful AR teacher, $p^*$. Formally, given an AR teacher model:

$$p^*(\boldsymbol{x}^{1:n}) = \prod_{i=1}^{n} p^*(x^i|\boldsymbol{x}^{<i}), \quad \text{where } \forall i, \ p^*(x^i = k|\boldsymbol{x}^{<i}) = \frac{\exp\left(f^*(\boldsymbol{x}^{<i})_k\right)}{\sum_{j=1}^{V} \exp\left(f^*(\boldsymbol{x}^{<i})_j\right)},$$

---

[1] Throughout the paper, we use bold symbols for vectors.

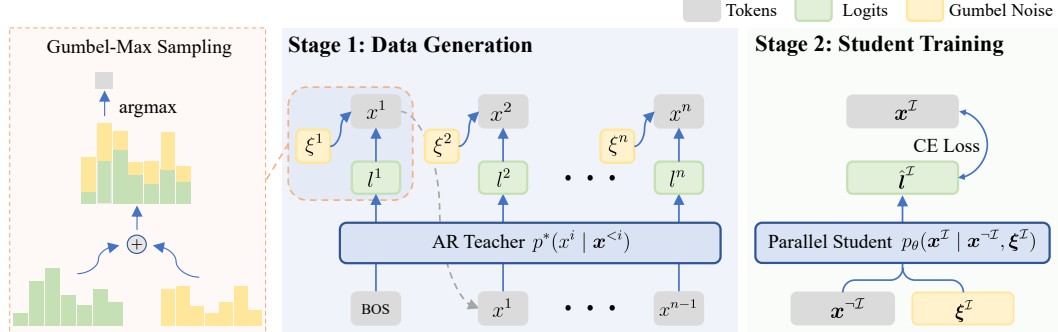

Figure 1: **A conceptual overview of Gumbel Distillation.** The distillation process consists of two steps: (1) **Data Generation:** An autoregressive teacher model's sampling process is combined with Gumbel noise to deterministically generate pairs of token sequences and their corresponding Gumbel noise. In practice, one could alternatively extract $\{\xi\}$ in parallel based on an offline training corpus (Section 4.1); (2) **Student Training:** A parallel student model is trained to predict a target subset of tokens, conditioned on both the context and the Gumbel noise from the teacher.

where $f^*(\cdot)_k$ denotes the $k$-th logit predicted by the teacher, we aim to solve the following problem:

*How can we train a parallel decoder $p_\theta$ to generate a subset of tokens $\boldsymbol{x}^{\mathcal{I}}$ simultaneously, while ensuring its output distribution $p_\theta(\boldsymbol{x}^{\mathcal{I}}|\boldsymbol{x}^{\neg\mathcal{I}})$ matches the teacher's distribution $p^*(\boldsymbol{x}^{\mathcal{I}}|\boldsymbol{x}^{\neg\mathcal{I}})$?*

However, this objective is notoriously difficult. A naive approach of using a neural network to directly model the conditional distribution $p_\theta(\boldsymbol{x}^{\mathcal{I}}|\boldsymbol{x}^{\neg\mathcal{I}})$ can easily become intractable. This is because the model would need to output a probability distribution over a discrete space of size $V^{|\mathcal{I}|}$, which grows exponentially with the number of tokens $|\mathcal{I}|$ generated in parallel.

To overcome these challenges, our key insight is to reframe the difficult distribution matching problem into a more manageable **supervised learning problem**. We achieve this by reformulating the teacher's stochastic sampling process as a fixed function of random noise. At each generation step, the AR model samples a token from a categorical distribution. The Gumbel-Max trick provides an equivalent formulation: adding random Gumbel noise to the logits and taking the argmax. This reveals a crucial link: for any sequence $\boldsymbol{x}^{1:n}$ generated by the teacher, there exists a corresponding sequence of Gumbel noise $\boldsymbol{\xi}^{1:n}$ that produces it, thereby creating an explicit mapping from noise to text. Consequently, instead of directly learning the joint distribution over a block of multiple tokens, the student's task is simplified to learning this deterministic function: $p_\theta(\boldsymbol{x}^{\mathcal{I}}|\boldsymbol{x}^{\neg\mathcal{I}}, \boldsymbol{\xi}^{\mathcal{I}})$.

## 4.1 GUMBEL DISTILLATION: FRAMEWORK

Building on the above intuition, we detail Gumbel Distillation as a two-stage framework (Figure 1). The process begins with data generation to produce training instances from the teacher, followed by student training, where the parallel decoder learns the noise-conditioned mapping.

**Stage 1: Data Generation**  In the first stage, we use the AR teacher $p^*$ to generate a complete pair consisting of a full token sequence $\boldsymbol{x}^{1:n}$ and its corresponding Gumbel noise sequence $\boldsymbol{\xi}^{1:n}$. This initial step is inherently sequential. From this complete pair, we then construct a training example for our parallel student based on its architecture. By selecting a subset of indices $\mathcal{I} \subseteq \{1, \ldots, n\}$ to be the prediction target, we partition the data into triplets $(\boldsymbol{x}^{\neg\mathcal{I}}, \boldsymbol{\xi}^{\mathcal{I}}, \boldsymbol{x}^{\mathcal{I}})$, where:

- $\boldsymbol{x}^{\neg\mathcal{I}}$ is the context, consisting of the tokens not in our target set.
- $\boldsymbol{\xi}^{\mathcal{I}}$ is the conditional Gumbel noise for the target positions.
- $\boldsymbol{x}^{\mathcal{I}}$ is the target token subset that the student learns to predict.

**Stage 2: Student Training**  Once the training data is prepared, the second stage involves training a parallel student decoder $p_\theta$ (e.g. masked diffusion model or multi-token-prediction model) to predict the target token subset $\boldsymbol{x}^{\mathcal{I}}$ simultaneously. The student model is conditioned on both the context $\boldsymbol{x}^{\neg\mathcal{I}}$ and the corresponding Gumbel noise for the target positions $\boldsymbol{\xi}^{\mathcal{I}}$, and the training objective is to maximize the conditional log-likelihood:

$$\mathcal{L} = -\mathbb{E}_{(\boldsymbol{x}^{\neg\mathcal{I}}, \boldsymbol{\xi}^{\mathcal{I}}, \boldsymbol{x}^{\mathcal{I}}) \sim \text{data}} \left[ \log p_\theta(\boldsymbol{x}^{\mathcal{I}}|\boldsymbol{x}^{\neg\mathcal{I}}, \boldsymbol{\xi}^{\mathcal{I}}) \right].$$

---

**Algorithm 1** Parallel Gumbel Sampling for a Sequence

---

**Require:** Ground-truth token sequence $x^{1:n} = (x^1, \ldots, x^n)$
**Require:** Sequence of logit vectors $\boldsymbol{l}^{1:n} = (\boldsymbol{l}^1, \ldots, \boldsymbol{l}^n)$, where each $\boldsymbol{l}^i \in \mathbb{R}^V$
**Ensure:** Posterior Gumbel sample sequence $\boldsymbol{\xi}^{1:n} = (\boldsymbol{\xi}^1, \ldots, \boldsymbol{\xi}^n)$, where each $\boldsymbol{\xi}^i \in \mathbb{R}^V$
 1: **for all** $i \in \{1, \ldots, n\}$ **in parallel do**
 2:     $\boldsymbol{p}^i \leftarrow \text{Softmax}(\boldsymbol{l}^i)$
 3:     Sample auxiliary noises: $\zeta_0 \sim \mathcal{G}(0, 1)$ and $\boldsymbol{\zeta} \in \mathbb{R}^V \sim \mathcal{G}(0, 1)^V$
 4:     $\boldsymbol{\xi}^i \leftarrow -\log \left( \exp(-\boldsymbol{\zeta}) + \boldsymbol{p}^i \exp(-\zeta_0) \right)$         $\triangleright$ Assigns to the full vector $\boldsymbol{\xi}^i$
 5:     $\xi^i_{x^i} \leftarrow \zeta_0 - \log p^i_{x^i}$         $\triangleright$ Overwrites a single scalar component at index $x^i$
 6: **end for**
 7: **return** $\boldsymbol{\xi}^{1:n}$

---

One crucial implementation detail in this framework is how we obtain the set of distillation targets $(\boldsymbol{\xi}^{1:n}, \boldsymbol{x}^{1:n})$ from the AR teacher. We propose two practical methods for this extraction process. The first arises naturally from the Gumbel-Max procedure, while the second approximately equivalent method offers significant computational advantages by allowing us to parallelize the computation.

**Sequential Gumbel Extraction**     The most straightforward approach is to perform sequential sampling from the teacher model $p^*$. For each position $i = 1, \ldots, n$, we first draw a Gumbel noise vector $\boldsymbol{\xi}^i \in \mathbb{R}^V$ and then recursively apply the Gumbel-Max trick to generate the next token $x^i$,

$$x^i = \arg\max_k \left( \xi^i_k + f^*(\boldsymbol{x}^{<i})_k \right), \quad \text{where} \quad \xi^i_k = -\log(-\log(u^i_k)), \quad u^i_k \sim \text{Uniform}(0, 1).$$

By repeating this autoregressive process for the entire sequence, we can generate pairs of full sequences $(\boldsymbol{\xi}^{1:n}, \boldsymbol{x}^{1:n})$, and the mapping from the noise $\boldsymbol{\xi}^{1:n}$ to the text $\boldsymbol{x}^{1:n}$ is deterministic given that the AR model is fixed and deterministic. While computationally intensive, this approach allows us to generate an unlimited amount of high-fidelity distillation data, which can then be partitioned into training triplets as described above.

**Parallel Gumbel Extraction**     While the sequential strategy is simple and robust, it requires many forward passes through the teacher model to generate data. As a highly efficient alternative, we can leverage an existing text corpus by assuming its data was drawn from the teacher's distribution $p^*$. Given a token sequence $\boldsymbol{x}^{1:n}$ from the corpus, we first perform a single forward pass with the teacher model to get the corresponding logits $\boldsymbol{l}^{1:n}$ The problem is then reframed as recovering the posterior distribution of the Gumbel noise $P(\boldsymbol{\xi}^{1:n} | \boldsymbol{x}^{1:n}, \boldsymbol{l}^{1:n})$ that would reconstruct the original text.

To this end, the following theorem provides a direct, analytical method for sampling from this posterior, enabling us to extract Gumbel noise for an entire sequence in parallel. Using Algorithm 1, we can sample the Gumbel noise vectors for every token in a sequence via one single forward pass through the teacher model, which significantly accelerates the data generation process. A detailed version of the theorem and its proof is available in Appendix B

**Theorem 4.1** (Gumbel-Max Posterior Sampling)**.** *Assume token $x$ is drawn from the softmax distribution of a logits vector $\boldsymbol{l} = (l_1, \ldots, l_V)$, using the Gumbel-Max trick with a Gumbel noise vector $\boldsymbol{\xi} = (\xi_1, \ldots, \xi_V)$, i.e. $x = \arg\max_k(l_k + \xi_k)$. A valid sample $\boldsymbol{\xi}$ from the posterior distribution $p(\boldsymbol{\xi} | X = x)$ that satisfies the constraint can be drawn by the procedure which we extend for sequences in Algorithm 1.*

### 4.2   Gumbel Distillation: Application

Our framework benefits from its simplicity and generality. By reformulating the learning problem as a supervised task conditioned on Gumbel noise, Gumbel Distillation can serve as a versatile, plug-and-play enhancement for existing parallel decoders. In this section, we show how Gumbel Distillation can be seamlessly integrated into two popular families of such models: Masked Diffusion Language Models and Multi-Token-Prediction Models.

**Conditioning Masked Diffusion Language Models**     Masked Diffusion Language Models (e.g., MDLM (Sahoo et al., 2024), BD3-LM (Arriola et al., 2025)) operate by training a model to recover the original text from a corrupted version, where masking is used for the corruption process. In our setup, given a text sequence where a subset of tokens $\boldsymbol{x}^{\mathcal{I}}$ has been replaced by `[MASK]`, the model

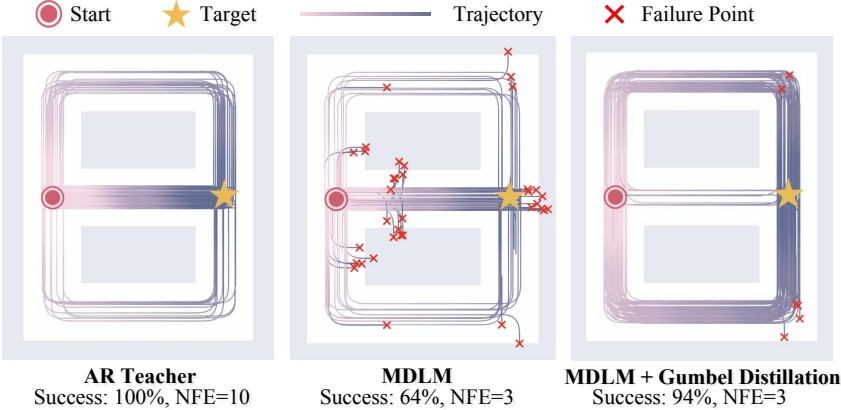

Figure 2: **A toy maze problem illustrating how Gumbel Distillation helps a student model learn a structured task.** The goal is to generate a valid path from a start (`<bos>`) to a target (`<eos>`) using a simple vocabulary (`up`, `down`, `left`, `right`). The figure visualizes 100 generated paths from each model, slightly jittered for visualization. The baseline MDLM frequently fails, producing incoherent paths, indicating its difficulty in modeling the joint distribution of the sequence. In contrast, our Gumbel-conditioned model successfully finds valid paths, closely matching the performance of the AR teacher but at a fraction of the generation cost (NFE = 3 steps vs. 10 steps).

learns to predict the original tokens $\boldsymbol{x}^{\mathcal{I}}$ based on the unmasked context $\boldsymbol{x}^{\neg\mathcal{I}}$. To apply Gumbel Distillation, we simply augment the model's input. Instead of predicting the masked tokens from the context alone, the student model $p_\theta$ is now also conditioned on the Gumbel noise $\boldsymbol{\xi}^{\mathcal{I}}$ corresponding to the target tokens. The learning objective becomes maximizing $p_\theta(\boldsymbol{x}^{\mathcal{I}} \mid \boldsymbol{x}^{\neg\mathcal{I}}, \boldsymbol{\xi}^{\mathcal{I}})$.

A crucial design choice is how to inject the Gumbel signal. We found it most effective to process the Gumbel noise for the masked tokens, $\boldsymbol{\xi}^{\mathcal{I}}$, into a vector on the model's vocabulary embedding space via a softmax normalization followed by a learned linear projection. Softmax function constrains the Gumbel values to $(0, 1)$, controlling for its large tails, while preserving the relative ranks in Gumbel that encode the teacher's sampling choice. These processed Gumbel embeddings then replace the standard `[MASK]` token embeddings in the input sequence. In this way, the uninformative `[MASK]` tokens are substituted with a rich "blueprint" from the teacher, guiding the student for prediction. Full implementation details can be found in Appendix C.1.

The visualization in Figure 2 provides a compelling toy example of this process, where we apply Gumbel Distillation to MDLM on a simple maze navigation task by defining the valid action sequence as the language. The Gumbel-conditioned MDLM improves greatly over the baseline to match the AR teacher's performance with a much lower NFE.

**Conditioning Multi-Token-Prediction Models** Multi-Token Prediction (MTP) models, such as Medusa (Cai et al., 2024), accelerate decoding by using multiple "heads" to simultaneously generate a block of future tokens. At a given generation step $i$ with $k$ heads, this block corresponds to the target indices $\mathcal{I} = \{i+1, \ldots, i+k\}$. To apply Gumbel Distillation, we train these heads to predict the target block $\boldsymbol{x}^{\mathcal{I}}$ conditioned on the preceding context $\boldsymbol{x}^{\neg\mathcal{I}}$ and the corresponding Gumbel noise blueprint $\boldsymbol{\xi}^{\mathcal{I}}$ from the teacher.

The Gumbel noise is processed into a conditioning vector and provided to each prediction head. This gives the heads direct guidance on the joint distribution of the token sequence, helping them overcome the standard conditional independence assumption and propose more coherent candidate blocks for verification. Full implementation details can be found in Appendix C.2.

## 5 EXPERIMENTS

In this section, we empirically validate Gumbel Distillation. We demonstrate its effectiveness and generality on Masked Diffusion Models (Section 5.1) and Multi-Token Prediction Models (Section 5.2). We conclude with ablation studies (Section 5.3) to justify our method's design.

Table 1: **Main results for unconditional text generation.** AR denotes an autoregressive model with the *same parameter as the student backbones* (used only as a quality reference); the distillation *teacher* is GPT-2-Large. Models are trained for 1M steps on LM1B (length 128) and OpenWebText (length 1024). Best non-AR scores are in bold.

| | LM1B | | OpenWebText | |
|---|---|---|---|---|
| | MAUVE ↑ | GenPPL ↓ | MAUVE ↑ | GenPPL ↓ |
| AR (student-size) | 0.465 | 36.42 | 0.691 | 14.10 |
| MDLM | 0.179 | 78.74 | 0.217 | 38.34 |
| MDLM + Gumbel Distillation | **0.264** | **67.64** | **0.282** | **34.33** |
| BD3-LM ($L' = 4$) | 0.193 | 56.98 | 0.251 | 26.40 |
| BD3-LM ($L' = 4$) + Gumbel Distillation | **0.291** | **46.06** | **0.304** | **24.37** |

Table 2: **Qualitative evaluation using Gemini-2.5-pro as an LLM judge on OpenWebText.** Scores are on a 1-10 scale (higher is better). Gumbel Distillation shows clear improvements across most dimensions, with the most significant and consistent gains observed in **Clarity** and **Factuality**.

| Model | Clarity | Grammaticality | Factuality | Style | Creativity |
|---|---|---|---|---|---|
| MDLM | 2.44 | 2.22 | 2.70 | 2.32 | 2.22 |
| MDLM + Gumbel Distillation | **2.86** (+17.2%) | **2.57** (+15.8%) | **3.31** (+22.6%) | **2.57** (+10.8%) | **2.36** (+6.3%) |
| BD3-LM | 2.89 | 2.95 | 3.21 | 3.34 | **2.75** |
| BD3-LM + Gumbel Distillation | **3.41** (+18.0%) | **3.22** (+9.2%) | **3.78** (+17.7%) | **3.35** (+0.2%) | 2.68 (-2.5%) |

## 5.1 GUMBEL DISTILLATION FOR MASKED DIFFUSION MODELS

**Experimental Setup.** To evaluate Gumbel Distillation, we integrate it into two state-of-the-art masked diffusion language model frameworks: **MDLM** and **BD3-LM**. For all distillation experiments, the autoregressive teacher is **GPT-2-Large**. For our main results, we use the efficient *parallel* Gumbel extraction method (Section 4.1) to generate distillation pairs online while iterating through the training data. We compare against two primary baselines: (i) an **AR model** with the same parameter budget as the student backbones (for a fair quality reference), and (ii) **students trained from scratch** (MDLM/BD3-LM) on raw text without distillation. Full implementation details are in Appendix C.1.

**Results on Unconditional Text Generation** We evaluate performance using two key metrics. **Generative Perplexity** (Gen. PPL, lower is better) measures token-level fluency. For a more holistic assessment of quality and diversity, we use the **MAUVE score (Liu et al., 2021)** (higher is better), which compares the generated text distribution to a human reference. As shown in Table 1, our method, Gumbel Distillation, consistently and significantly improves the performance of the baseline models in unconditional text generation. For instance, when applied to MDLM on Open-WebText, Gumbel Distillation reduces generative perplexity by 10.5% and boosts the MAUVE score by 30.0%. These substantial gains are consistent across other results on BD3-LM, demonstrating our approach's general effectiveness.

**LLM-based Evaluation** To complement our statistical metrics with a qualitative assessment, we used Gemini-2.5-pro (Comanici et al., 2025) to score the text samples generated by models trained on OWT. As shown in Table 2, we use Gemini-2.5-pro to judge all models on five dimensions: clarity, grammaticality, factuality, style, and creativity. Full details on the evaluation prompt and setup can be found in Appendix D.3. The results are consistent with our earlier findings using Gen. PPL and MAUVE score, confirming that Gumbel Distillation leads to a clear improvement in perceived generation quality. For both MDLM and BD3-LM, we see consistent gains across all dimensions, with particularly strong improvements in Factuality and Clarity. This suggests that the Gumbel conditioning helps the student model generate more coherent and grounded text.

**Performance vs. Number of Sampling Steps** In Figure 3, we plot MAUVE score and generative perplexity against the number of inference steps (NFE), using the efficient ancestral sampler from Sahoo et al. (2024) for all models. The results show two key advantages. First, at any given NFE, our models achieve a stronger perplexity than the baselines. Second, our models reach a high level of quality in far fewer steps compared to the baseline models. This efficiency gain suggests that

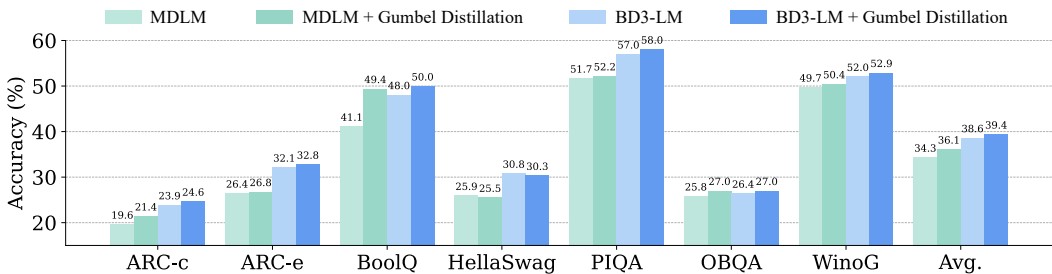

Figure 3: **Generative Perplexity and MAUVE Score vs. Number of Function Evaluations (NFE) of MDLM on LM1B and OpenWebText**. Our method (Gumbel Distillation) consistently outperforms the baselines, achieving lower perplexity for a given number of evaluations.

Figure 4: **Zero-shot performance on common-sense reasoning and question answering benchmarks.** Our Gumbel Distillation consistently improves accuracy over the baseline models across a suite of eight tasks. All scores are percentages (%).

the Gumbel "blueprint" provides a more direct signal, enabling the student model to converge on a high-quality output during the iterative generation process.

**Zero-shot Benchmark Performance** We evaluate the language understanding capabilities of the models trained on OWT on a wide range of question answering tasks. Following von Rütte et al. (2025), our selected benchmarks include ARC-e and ARC-c (Clark et al., 2018), BoolQ(Clark et al., 2019), HellaSwag (Zellers et al., 2019), PIQA (Bisk et al., 2020), OpenBookQA (Mihaylov et al., 2018) and Winogrande (Sakaguchi et al., 2021). The results, presented in Figure 4, show that Gumbel Distillation effectively transfers the reasoning capabilities of the AR teacher to the parallel student models. For instance, applying our method to MDLM improves its average accuracy across all benchmarks from 34.3% to 36.1%. We observe consistent gains across the board, with a particularly notable improvement in BoolQ. Similar improvements are seen when applying our method to BD3-LM (from 38.6% to 39.4%), demonstrating the robustness of our approach. This demonstrates that the student not only mimics the teacher's fluency but also inherits its knowledge and common-sense reasoning abilities, substantially improving the performance on these challenging tasks.

## 5.2 GUMBEL DISTILLATION FOR MULTI-TOKEN PREDICTION

We apply Gumbel Distillation to a MTP framework based on Medusa (Cai et al., 2024). This setup provides a direct, single-step test of a model's ability to capture the joint distribution of a sequence.

**Experimental Setup.** We follow the Medusa-1 setup, using a frozen GPT-2-Small backbone and training parallel MTP proposal heads (offsets 1–3, head 0 uses the frozen base LM head) on OpenWebText. The frozen backbone serves as the verifier, and we apply Gumbel Distillation in a self-distillation setup where the backbone also provides the teacher logits. To test scalability, we additionally apply Gumbel Distillation to Medusa heads on a frozen Vicuna-7B backbone. For all experiments, we use typical acceptance for speculative decoding and report the conditional acceptance rate of each head: a token proposed by a later head is only verified if all preceding tokens in the block have already been accepted. Full implementation details are in Appendix C.2.

**Results and Analysis** The results in Table 3 shows that Gumbel conditioning consistently improves per-head acceptance rates and increases the average number of accepted tokens per decoding step. On GPT-2-Small, the relative gains grow with head index, from +4.5% on Head 1 to +22.0% on Head 3. Crucially, these gains persist when scaling to Vicuna-7B, from +8.9% on Head 1 to +37.6% on Head 3 This trend provides strong evidence that Gumbel Distillation helps the MTP

Table 3: **Per-position acceptance rates of Medusa MTP heads.** We report conditional acceptance rates under typical acceptance. $\Delta_{\mathrm{rel}}\%$ denotes relative improvement over the baseline. Top: frozen GPT-2-Small trained on OpenWebText (self-distillation). Bottom: scaling to frozen Vicuna-7B trained on 60k ShareGPT samples (Medusa recipe; Head 0 uses the frozen LM head).

| | Baseline | + Gumbel Distillation | $\Delta_{\mathrm{rel}}\%$ |
|---|---|---|---|
| **GPT-2-Small backbone (OpenWebText)** | | | |
| Head 0 (NTP) | 1.000 | 1.000 | - |
| Head 1 | 0.445 | 0.465 | (+4.5%) |
| Head 2 | 0.310 | 0.344 | (+11.0%) |
| Head 3 | 0.264 | 0.322 | (+22.0%) |
| Accepted Length | 1.609 | 1.691 | (+5.1%) |
| **Vicuna-7B backbone (ShareGPT)** | | | |
| Head 0 (LM head) | 1.000 | 1.000 | – |
| Head 1 | 0.545 | 0.593 | (+8.9%) |
| Head 2 | 0.302 | 0.388 | (+28.1%) |
| Head 3 | 0.214 | 0.294 | (+37.6%) |
| Accepted Length | 1.745 | 1.891 | (+8.4%) |

heads learn the stronger sequential dependencies required to predict tokens further into the future, directly addressing the core challenge of modeling a joint distribution.

## 5.3 ABLATION STUDIES

Table 4: **Comparison to outcome-based distillation baselines and APD on MDLM (LM1B).** Token-level and sequence-level KD are outcome-based AR to NAR distillation baselines; APD is applied only at inference time. Best MAUVE and best GenPPL are in bold.

| Method | MAUVE ↑ | GenPPL ↓ |
|---|---|---|
| MDLM | 0.179 | 78.74 |
|   + Token-level KD | 0.166 | 95.88 |
|   + Sequence-level KD | 0.169 | 99.48 |
|   + APD | 0.203 | 57.61 |
|   + Gumbel Distillation | **0.264** | 67.64 |
|   + Gumbel Distillation + APD | 0.255 | **49.28** |

**Additional Comparison to Classical Distillation Baselines and APD.** We compare Gumbel Distillation to *outcome-based* AR to Non-AR distillation baselines as well as an inference-time acceleration method. Specifically, we consider two forms of Knowledge Distillation (KD): (i) token-level KD (Hinton et al., 2015), which matches the teacher's per-position output distribution (e.g., via KL divergence between teacher and student logits), and (ii) sequence-level KD (Kim & Rush, 2016), which trains the student on sequences generated by the AR teacher. We also test Adaptive Parallel Decoding (APD) (Israel et al., 2025), an inference-time procedure that uses a lightweight AR verifier to adaptively accept the longest high-quality prefix from diffusion proposals. Unlike KD and Gumbel Distillation, APD *does not modify training*: it is orthogonal and can be applied on top of any trained model, including our Gumbel-distilled models. All methods are evaluated on the same MDLM backbone and LM1B protocol as above. Results are shown in Table 4. Token-level KD primarily aligns per-position marginals under the student's conditional-independence assumptions within a parallel block, which does not enforce coherent joint decisions and can even conflict with the original masked-diffusion training objective, leading to worse fluency; Sequence-level KD replaces the training distribution with teacher-generated text, which can reduce diversity and encourage mode collapse, yielding worse MAUVE/GenPPL in our setting. In comparison, Applying APD on top of a Gumbel-distilled MDLM yields the best GenPPL, with a modest reduction in MAUVE.

**Impact of Sequential vs. Parallel Gumbel Extraction** As previously discussed in Section 4.1, we proposed two methods for generating the (`noise`, `text`) pairs for distillation: Sequential Extraction, which generates new text via ancestral sampling from the teacher, and Parallel Extraction,

Table 5: **Consolidated ablation study using MDLM on LM1B.** Parallel Gumbel Extraction is superior to the sequential method. The specific Gumbel distribution is critical, as replacing it with Gaussian noise degrades performance, and Uniform noise leads to mode collapse.

| Parallel Gumbel Extraction | Sequential Gumbel Extraction | Gumbel Noise | Gaussian Noise | Uniform Noise | MAUVE Score($\uparrow$) | Gen. PPL ($\downarrow$) |
|:---:|:---:|:---:|:---:|:---:|:---:|:---:|
| $\checkmark$ | | $\checkmark$ | | | **0.264** | **67.64** |
| | $\checkmark$ | $\checkmark$ | | | 0.189 | 86.38 |
| $\checkmark$ | | | $\checkmark$ | | 0.242 | 81.43 |
| $\checkmark$ | | | | $\checkmark$ | 0.097 | - |

which recovers the posterior Gumbel noise from an existing text corpus. Although we have provided theoretical support for the equivalence of these two methods, it still remains to be validated empirically whether Parallel Extraction works as well as Sequential Extraction in practice. Therefore, we conduct ablation experiments by obtaining a dataset with the same size as LM1B via the sequential method, and then train MDLM with and without Gumbel Distillation on this synthetic dataset.

Table 5 shows that Parallel Gumbel Extraction yields better performance than its sequential counterpart, reducing generative perplexity from 86.38 to 67.64. This result may seem counterintuitive, as sequential sampling generates data directly from the teacher's distribution. We suppose that it is susceptible to the teacher model's own errors and biases. If the teacher generates repetitive or low-quality text, the student will be trained to replicate these imperfections. In contrast, Parallel Extraction leverages a high-quality, diverse text corpus as the ground truth for $x^{1:n}$. It then uses the teacher only to infer the posterior Gumbel noise $\xi^{1:n}$ that would have led to the high-quality text.

**Impact of Gumbel vs. Other Noise Sources**   Our method's core principle is the deterministic link between Gumbel noise and token probabilities established by the Gumbel-Max trick. To check whether the specific properties of the Gumbel distribution are critical or harmful, we ablate our approach by replacing it with two other noise sources. (1) **Gaussian Noise:** A standard random signal commonly used in deep learning, which we transform Gumbel into and then feed as input; (2) **Uniform Noise:** A particularly important baseline, as it is the precursor to Gumbel noise in the inverse transform sampling procedure ($\xi = -\log(-\log(u))$). This directly tests if the specific transformation to the Gumbel distribution is necessary, or if any simple random variable is sufficient.

As shown in Table 5, both alternatives fail to match the performance of Gumbel noise. Using Gaussian noise as the conditional signal significantly degrades performance, resulting in worse Gen. PPL than the naive baseline. Using Uniform noise leads to training instability and mode collapse, where the model generates extremely low-diversity text. The choice of Gumbel noise is fundamental to creating a structured "blueprint" that the student model can effectively learn from.

## 6 CONCLUSION

In this work, we introduced Gumbel Distillation, a novel distillation framework designed to help parallel decoders overcome their struggle to model the joint distribution of token sequences. By using the Gumbel-Max trick to create a deterministic "blueprint" from an autoregressive teacher's sampling process, Gumbel Distillation reframes the intractable distribution matching problem into a simple supervised task. Our experiments demonstrate that it is a versatile, plug-and-play module that improves the quality, efficiency, and reasoning capabilities of diverse architectures like Mask Diffusion Models and MTPs. Future work could involve scaling this technique to larger foundation models and exploring the latent Gumbel space as a new avenue for controllable text generation.

**Limitation.**   One limitation of this work is that as vocabulary size $V$ grows, the dimensionality of the Gumbel noise vector grows proportionally. In our implementation this adds a single projection of cost $\mathcal{O}(VH)$ (applied once per sequence/block) and does not scale with model depth, so the *relative* parameter/compute overhead typically decreases as the transformer backbone scales (Appendix B.5). Still, this can lead to higher computational costs and more challenges in high-dimensional spaces. Future work could further mitigate this cost via structured or low-rank representations of the Gumbel noise. Despite this, our work paves the way for developing highly efficient generative models without compromising on quality, making powerful generative AI more accessible for real-world applications.

ACKNOWLEDGMENTS

We thank Chaochao Yan for helpful discussions and feedback. We also thank members of the UT Machine Learning Lab for comments and for computational support. This work was supported in part by the Institute for Foundations of Machine Learning (IFML), the Office of Naval Research (ONR) under Grant No. N00014-25-1-2354 and a grant from Google.

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

## A   USE OF LARGE LANGUAGE MODELS

Regarding paper writing, we used LLM only for text polishing and grammar correction during manuscript preparation. No LLMs were involved in the conception or design of the method, experiments, or analysis. All technical content, results, and conclusions have been independently verified and validated by the authors. Apart from this, we also used LLM as an evaluation tool in our experiments to judge the generation quality of text samples.

## B   ON THE VALIDITY OF THE POSTERIOR GUMBEL SAMPLING METHOD

In this section we give a formal proof of Theorem 4.1. Similar techniques on the posterior Gumbel distribution have been explored in prior works such as Kool et al. (2019), and we use some established findings from Maddison et al. (2014). Before proceeding with the proof, we first restate the full theorem from the main text.

**Theorem** (Posterior of the Gumbel-Max Process). *Assume $X \in [V]$ is sampled from a set of logits $\boldsymbol{l} = (l_1, \ldots, l_V) \in \mathbb{R}^V$ using the Gumbel-Max trick, such that $X = \arg\max_{k \in [V]} (l_k + \xi_k)$, where $\forall k, \xi_k \sim \mathcal{G}(0,1)$ are i.i.d. standard Gumbel noises. Let $p_k = \frac{\exp(l_k)}{\sum_j \exp(l_j)}$ be the softmax probability for each category $k \in [V]$ and $\boldsymbol{\xi} = (\xi_1, \ldots, \xi_V)$. The posterior probability density function over $\boldsymbol{\xi}$ conditioned on the outcome $X = x$ is given by:*

$$p(\boldsymbol{\xi}|X = x) = \frac{1}{p_x} \cdot \mathbf{1}_{\{l_x + \xi_x \geq l_k + \xi_k, \forall k \in [V]\}} \cdot \prod_{k=1}^{V} e^{-\xi_k - e^{-\xi_k}} \tag{1}$$

*where $\mathbf{1}_{\{.\}}$ is the indicator function.*

*Furthermore, a sample $\boldsymbol{\xi}$ can be drawn from this posterior by first sampling auxiliary i.i.d. variables $\zeta_0, \ldots, \zeta_V \sim \mathcal{G}(0,1)$ and then applying the construction:*

$$\xi'_k = \begin{cases} \zeta_0 - \log p_x & \text{if } k = x \\ -\log\left(\frac{\exp(-\zeta_k)}{p_k} + \exp(-\zeta_0)\right) & \text{if } k \neq x \end{cases} \tag{2}$$

*This constructed sample $\boldsymbol{\xi}'$ is guaranteed to have the following two properties:*

*1. (Condition satisfaction) The argmax of the perturbed logits matches the conditioned outcome.*

$$\arg\max_{k \in [V]} (l_k + \xi'_k) = x$$

*2. (Marginal Preservation) Each component of the sample is marginally a standard Gumbel.*

$$\forall k \in [V], \quad \xi'_k \sim \mathcal{G}(0,1)$$

### B.1   PROOF OF PROPERTY 1

*Proof.* We show that for any $k \neq x$, the inequality $l_x + \xi'_x > l_k + \xi'_k$ holds. By substituting the construction formulas and the identity $l_x - l_k = \log p_x - \log p_k$, the inequality becomes:

$$(\log p_x - \log p_k) + \zeta_0 - \log p_x > -\log\left(\frac{\exp(-\zeta_k)}{p_k} + \exp(-\zeta_0)\right)$$

$$-\log p_k + \zeta_0 > -\log\left(\frac{\exp(-\zeta_k)}{p_k} + \exp(-\zeta_0)\right)$$

$$\zeta_0 > \log\left(\frac{p_k}{\frac{\exp(-\zeta_k)}{p_k} + \exp(-\zeta_0)}\right)$$

Exponentiating both sides and rearranging yields:

$$\frac{e^{\zeta_0 - \zeta_k}}{p_k} + 1 > p_k$$

This inequality holds because the left-hand side is always strictly greater than 1, while $p_k \in [0, 1]$.
$\square$

### B.2  PROOF OF PROPERTY 2

We prove that each component of the constructed sample $\xi'_k$ is marginally distributed as a standard Gumbel, $\mathcal{G}(0,1)$. We handle the winning ($k = x$) and losing variables ($k \neq x$) separately.

CASE 1: THE WINNING VARIABLE ($\xi_x$)

*Proof.* The proof for the winning variable's marginal distribution relies on the **Posterior of the Maximum** property of the Gumbel distribution. This property states that the maximum score, $M = \max_{k \in [V]}(l_k + \xi_k)$, conditioned on the winning category being $x$, follows a Gumbel distribution centered at the winning logit $l_x$:

$$p(M \mid \arg\max_{k \in [V]}(l_k + \xi_k) = x) \sim \mathcal{G}(l_x, 1)$$

The posterior construction is equivalent to a top-down sampling process where a sample of the maximum score, call it $m^*$, is first drawn from this posterior distribution. Thus, $m^* \sim \mathcal{G}(l_x, 1)$.

By definition, $m^*$ must be equal to the score of the winning category from our constructed sample:

$$m^* = l_x + \xi'_x$$

By rearranging the equation, we can express the constructed Gumbel noise $\xi_x$ in terms of $m^*$:

$$\xi_x = m^* - l_x$$

Since $m^*$ is a randomly sampled from $\mathcal{G}(l_x, 1)$, subtracting the location parameter $l_x$ simply shifts the distribution's location to zero. Therefore, the resulting random variable $\xi_x$ follows:

$$\xi_x \sim \mathcal{G}(l_x - l_x, 1) = \mathcal{G}(0, 1)$$

Thus, the marginal distribution of any winning variable $\xi_x$ is the standard Gumbel distribution. $\square$

CASE 2: THE LOSING VARIABLE ($\xi_k, k \neq x$)

*Proof.* The proof for the losing variables is more involved. We show that the cumulative distribution function (CDF) of $\xi_k$ for any $k \neq x$ matches the standard Gumbel CDF, $F(t) = e^{-e^{-t}}$.

The construction formula for a losing variable is:

$$\xi_k = -\log\left(\frac{\exp(-\zeta_k)}{p_k} + \exp(-\zeta_0)\right)$$

The proof proceeds with a change of variables. We use the property that if $\zeta \sim \mathcal{G}(0,1)$, then its transformation $U = e^{-\zeta}$ follows the standard Exponential distribution, $U \sim \text{Exponential}(1)$. Let $U_k = e^{-\zeta_k}$ and $U_0 = e^{-\zeta_0}$. Both are i.i.d. samples from Exponential(1).

The CDF, $P(\xi_k \leq t)$, is equivalent to the probability $P\left(\frac{U_k}{p_k} + U_0 \geq e^{-t}\right)$. This probability is found by integrating the joint PDF of $U_k$ and $U_0$, which is $f(u_k, u_0) = e^{-u_k}e^{-u_0}$ for $u_k, u_0 \geq 0$, over the region satisfying the inequality. Letting $c = e^{-t}$, the integral is:

$$P(\xi_k \leq t) = \int_0^\infty \int_0^\infty \mathbf{1}_{\left\{\frac{u_k}{p_k} + u_0 \geq c\right\}} e^{-u_k} e^{-u_0}\, du_k\, du_0$$

While we omit the full integration here for brevity, it simplifies to the CDF.

$$P(\xi_k \leq t) = e^{-c} = e^{-e^{-t}}$$

Thus, the marginal distribution of any losing variable $\xi_k$ is the standard Gumbel distribution. $\square$

**Theorem** (Posterior of the Gumbel-Max Process). *Assume $X \in [V]$ is sampled from a set of logits $\boldsymbol{l} = (l_1, \ldots, l_V) \in \mathbb{R}^V$ using the Gumbel-Max trick, such that $X = \arg\max_{k \in [V]}(l_k + \xi_k)$, where $\forall k, \xi_k \sim \mathcal{G}(0,1)$ are i.i.d. standard Gumbel noises. Let $p_k = \frac{\exp(l_k)}{\sum_j \exp(l_j)}$ be the softmax probability*

*for each category $k \in [V]$ and $\boldsymbol{\xi} = (\xi_1, \ldots, \xi_V)$. The posterior probability density function over $\boldsymbol{\xi}$ conditioned on the outcome $X = x$ is given by:*

$$p(\boldsymbol{\xi}|X = x) = \frac{1}{p_x} \cdot \mathbf{1}_{\{l_x + \xi_x \geq l_k + \xi_k, \forall k \in [V]\}} \cdot \prod_{k=1}^{V} e^{-\xi_k - e^{-\xi_k}} \tag{3}$$

*where $\mathbf{1}_{\{\cdot\}}$ is the indicator function.*

*Furthermore, a sample $\boldsymbol{\xi}'$ can be drawn from this posterior by first sampling auxiliary i.i.d. variables $\zeta_0, \ldots, \zeta_V \sim \mathcal{G}(0, 1)$ and then applying the construction:*

$$\xi_k' = \begin{cases} \zeta_0 - \log p_x & \text{if } k = x \\ -\log\left(\exp(-\zeta_k) + p_k \exp(-\zeta_0)\right) & \text{if } k \neq x \end{cases} \tag{4}$$

*This constructed sample $\boldsymbol{\xi}'$ is guaranteed to have the following two properties:*

*1. (Condition satisfaction) The argmax of the perturbed logits matches the conditioned outcome.*

$$\arg\max_{k \in [V]}(l_k + \xi_k') = x$$

*2. (Marginal Preservation) The unconditional marginal distribution of the constructed sample matches the standard Gumbel prior.*

$$\forall k \in [V], \quad \xi_k' \sim \mathcal{G}(0, 1)$$

### B.3 PROOF OF PROPERTY 1

*Proof.* We show that for any $k \neq x$, the perturbed score of the winning category strictly dominates, i.e., $l_x + \xi_x' > l_k + \xi_k'$. Substituting the construction formula for the winning variable ($k = x$), we have:

$$l_x + \xi_x' = l_x + \zeta_0 - \log p_x = \zeta_0 + \log Z$$

where we applied the softmax identity $\log p_x = l_x - \log Z$.

For any losing variable ($k \neq x$), since $\zeta_k \sim \mathcal{G}(0, 1)$, its exponential is strictly positive ($\exp(-\zeta_k) > 0$). This yields the strict inequality:

$$\exp(-\zeta_k) + p_k \exp(-\zeta_0) > p_k \exp(-\zeta_0)$$

Because the negative logarithm is a strictly decreasing function, applying $-\log(\cdot)$ to both sides reverses the inequality:

$$\xi_k' = -\log\left(\exp(-\zeta_k) + p_k \exp(-\zeta_0)\right)$$
$$< -\log\left(p_k \exp(-\zeta_0)\right) = \zeta_0 - \log p_k$$

Adding $l_k$ to both sides and applying $\log p_k = l_k - \log Z$, we obtain the score for class $k$:

$$l_k + \xi_k' < l_k + \zeta_0 - \log p_k = \zeta_0 + \log Z$$

Comparing the two scores, $l_x + \xi_x' = \zeta_0 + \log Z > l_k + \xi_k'$. Thus, $\arg\max_{k \in [V]}(l_k + \xi_k') = x$.

$\square$

### B.4 PROOF OF PROPERTY 2

*Proof.* To rigorously prove Marginal Preservation, we must show that the unconditional distribution of each constructed component $\xi_k'$ is exactly the standard Gumbel distribution $\mathcal{G}(0, 1)$.

Instead of working directly with the Gumbel PDF, we apply the transformation $E_k = \exp(-\xi_k')$. Since $\xi \sim \mathcal{G}(0, 1) \iff \exp(-\xi) \sim \text{Exp}(1)$, our objective is equivalent to proving that the unconditional survival function of $E_k$ is $P(E_k > t) = e^{-t}$ for any $t \geq 0$.

Let $U_0 = \exp(-\zeta_0)$ and $U_k = \exp(-\zeta_k)$. Since $\zeta_0, \zeta_k \sim \mathcal{G}(0, 1)$ are independent, $U_0$ and $U_k$ are independent standard exponential variables, $U_0, U_k \sim \text{Exp}(1)$.

By the Law of Total Probability, we condition on whether class $k$ is the winning category ($X = k$, which occurs with probability $p_k$) or a losing category ($X \neq k$, which occurs with probability $1 - p_k$):

$$P(E_k > t) = p_k \cdot P(E_k > t \mid X = k) + (1 - p_k) \cdot P(E_k > t \mid X \neq k) \tag{5}$$

**Case 1:** $X = k$ **(The Winning Category)** From the construction formula, $\xi'_k = \zeta_0 - \log p_k$. Transforming to the exponential domain:

$$E_k = \exp(-(\zeta_0 - \log p_k)) = p_k \exp(-\zeta_0) = p_k U_0$$

The conditional survival probability is:

$$P(E_k > t \mid X = k) = P(p_k U_0 > t) = P\left(U_0 > \frac{t}{p_k}\right) = \exp\left(-\frac{t}{p_k}\right) \tag{6}$$

**Case 2:** $X \neq k$ **(The Losing Category)** From the construction formula, $\xi'_k = -\log(\exp(-\zeta_k) + p_k \exp(-\zeta_0))$. Transforming to the exponential domain:

$$E_k = \exp(-\zeta_k) + p_k \exp(-\zeta_0) = U_k + p_k U_0$$

The conditional survival probability is $P(U_k + p_k U_0 > t)$. Since $U_k, U_0 \sim \mathrm{Exp}(1)$ are independent, their joint PDF is $f(u_k, u_0) = e^{-u_k} e^{-u_0}$. We integrate this over the region $u_k + p_k u_0 > t$:

$$\begin{aligned}
P(U_k + p_k U_0 > t) &= \int_0^\infty du_0 \, e^{-u_0} \int_{\max(0, t - p_k u_0)}^\infty du_k \, e^{-u_k} \\
&= \int_0^{t/p_k} e^{-u_0} e^{-(t - p_k u_0)} \, du_0 + \int_{t/p_k}^\infty e^{-u_0} (1) \, du_0 \\
&= e^{-t} \int_0^{t/p_k} e^{-u_0(1 - p_k)} \, du_0 + \exp\left(-\frac{t}{p_k}\right) \\
&= e^{-t} \left[\frac{1}{-(1 - p_k)} e^{-u_0(1 - p_k)}\right]_0^{t/p_k} + \exp\left(-\frac{t}{p_k}\right) \\
&= \frac{e^{-t}}{1 - p_k} \left(1 - \exp\left(-\frac{t}{p_k}(1 - p_k)\right)\right) + \exp\left(-\frac{t}{p_k}\right) \\
&= \frac{e^{-t}}{1 - p_k} - \frac{\exp(-t/p_k)}{1 - p_k} + \exp\left(-\frac{t}{p_k}\right) \\
&= \frac{e^{-t} - p_k \exp(-t/p_k)}{1 - p_k}
\end{aligned}$$

**Combining the Cases:** Substitute the results from Case 1 and Case 2 back into the Law of Total Probability:

$$\begin{aligned}
P(E_k > t) &= p_k \exp\left(-\frac{t}{p_k}\right) + (1 - p_k) \left(\frac{e^{-t} - p_k \exp(-t/p_k)}{1 - p_k}\right) \\
&= p_k \exp\left(-\frac{t}{p_k}\right) + e^{-t} - p_k \exp\left(-\frac{t}{p_k}\right) \\
&= e^{-t}
\end{aligned}$$

The result $P(E_k > t) = e^{-t}$ is exactly the survival function of the standard Exponential distribution. Therefore, $E_k \sim \mathrm{Exp}(1)$, which implies $\xi'_k = -\log E_k \sim \mathcal{G}(0, 1)$. This strictly completes the proof of marginal preservation. $\qquad\square$

### B.5 COMPLEXITY OF GUMBEL CONDITIONING

Gumbel Distillation conditions the student on a Gumbel vector $\xi \in \mathbb{R}^V$ via a single projection $W_\xi \in \mathbb{R}^{H \times V}$ into the model hidden dimension $H$. This introduces $VH$ additional parameters and one extra matrix multiplication per block, independent of the number of transformer layers. In contrast, the backbone transformer cost scales roughly with $\mathcal{O}(LH^2)$, so the relative overhead typically decreases as $L$ and $H$ increase.

Table 6: **Parameter overhead of the Gumbel projection matrix** ($W_\xi \in \mathbb{R}^{H \times V}$). Overhead is $VH$/TotalParams. Large vocabularies increase overhead, while larger backbones reduce it.

| Model scale | Layers $L$ | Hidden $H$ | Vocab $V$ | Added params $VH$ | Overhead |
|---|---|---|---|---|---|
| GPT-2 Small | 12 | 768 | 50,257 | 38.6M | 33.0% |
| GPT-2 Large | 36 | 1,280 | 50,257 | 64.3M | 8.3% |
| LLaMA-7B | 32 | 4,096 | 32,000 | 131.1M | 1.9% |
| LLaMA-13B | 40 | 5,120 | 32,000 | 163.8M | 1.3% |
| Qwen-7B | 32 | 4,096 | 152,000 | 622.6M | 8.9% |

## C EXPERIMENTAL DETAILS

### C.1 MASKED DIFFUSION LANGUAGE MODELS

**MDLM and BD3-LM**   Text diffusion models (Austin et al., 2021; Li et al., 2022) extend diffusion from visual domains to language modeling. The design is to reverse a forward process $q$ that gradually corrupts clean text into noise, which can use continuous or discrete representations. A denoising model $p_\theta$ is trained to recover the original text $x$ from a noisy version $x_t$ where $t \in [0, 1]$ is the noise level. The learning objective is typically the minimization of the Negative Evidence Lower Bound (NELBO), which can be expressed in a simplified form as:

$$-\log p_\theta(x) \leq \mathcal{L}_{\text{NELBO}}(x; \theta) = \mathbb{E}_{t, x_t \sim q(\cdot|x)} \left[ -w(t) \cdot \log p_\theta(x \mid x_t) \right], \tag{7}$$

A prominent family of text diffusion models uses masking as the corruption process. Masked Diffusion Language Model (MDLM) (Sahoo et al., 2024) is a simple yet effective architecture that learns to predict original tokens given a partially masked sequence. However, MDLM is designed only for fixed-length generation. To address this, Block Discrete Denoising Diffusion Language Model (BD3-LM) (Arriola et al., 2025) extends the idea by interpolating between diffusion and AR models. It generates text autoregressively over blocks of tokens, but within each block, it uses a diffusion process identical to that of MDLM. This hybrid approach enables variable-length generation. Because the core objective within a block is the same for both models, we use the simpler MDLM architecture for illustration in Figure 5, but use the full BD3-LM objective for our derivations below.

The BD3-LM objective factorizes the loss across $B$ blocks as follows:

$$\mathcal{L}_{\text{NELBO}}(x^{1:n}; \theta) = \sum_{b=1}^{B} \mathcal{L}_{\text{NELBO}}(x^{(b-1)c+1:bc} \mid x^{\leq (b-1)c}; \theta). \tag{8}$$

Here, the loss for each block is conditioned on the clean, previously generated blocks.

We adapt this block-wise objective to be conditioned on our Gumbel signal. The overall task is to learn the conditional distribution $p_\theta(x^{1:n}|\boldsymbol{\xi}^{1:n})$, and we achieve this by modifying the BD3-LM objective from Equation 8, adding the corresponding slice of the Gumbel sequence as a condition for each block. This results in a conditional NELBO:

$$\mathcal{L}_{\text{NELBO}}(x^{1:n}|\boldsymbol{\xi}^{1:n}; \theta) = \sum_{b=1}^{B} \mathcal{L}_{\text{NELBO}}(x^{(b-1)c+1:bc} \mid x^{\leq (b-1)c}, \boldsymbol{\xi}^{(b-1)c+1:bc}; \theta). \tag{9}$$

As mentioned in Section 4.2, a crucial implementation detail is how the Gumbel signal $\boldsymbol{\xi}^{1:n}$ is injected into the model. In practice, we process the raw Gumbel noise for each block, $\boldsymbol{\xi}^b$, into a dense vector on the embedding space, $g(\boldsymbol{\xi}^b)$, by first performing a softmax operation for normalization, followed by a learned linear projection. Then we use to replace the embeddings of [MASK] tokens in the noised input. Formally, let $E(x_t^b)$ be the initial token embeddings for a noised block $b$, let $M^b$ be a binary mask where $M_i^b = 1$ if the $i$-th token is [MASK] and 0 otherwise, and let $W_{proj}$ be the weight matrix of the linear layer. Then, the final input to the model's transformer layers, $\tilde{E}(x_t^b)$, is:

$$\tilde{E}^b = (1 - M^b) \odot E(x_t^b) + M^b \odot g(\xi^b) \quad \text{where} \quad g(\xi^b) = \text{softmax}(\xi^b) W_{proj}. \tag{10}$$

Here $\odot$ denotes element-wise multiplication. This design substitutes the uninformative [MASK] embeddings with Gumbel signals, which serve as a "blueprint" for what to predict at those positions. In Figure 5, we give an illustration of how Gumbel Distillation is integrated into MDLM.

**Datasets** Following Sahoo et al. (2024); Arriola et al. (2025), we conduct our experiments on two datasets, The One Billion Word Dataset; **LM1B** (Chelba et al., 2014) and OpenWebText; **OWT** (Gokaslan & Cohen, 2019). We use a context length of 128 for models trained on LM1B and 1024 for models trained on OWT. For both datasets, we use the `GPT-2` tokenizer to keep consistent with our selected teacher model. For LM1B, sequences were concatenated and wrapped to a maximum length of 128. For OWT, documents were concatenated and chunked into sequences of 1,024 tokens, with an [eos] token used as a separator between documents. Since OWT does not have an official validation split, we created one by holding out the last 100,000 documents.

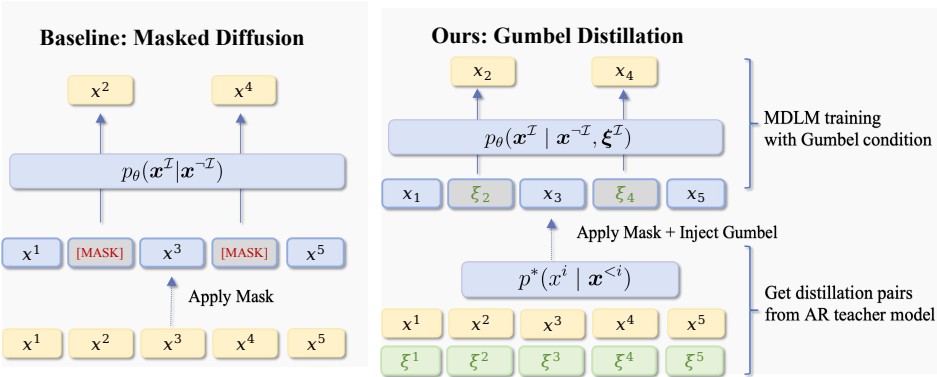

Figure 5: Gumbel Distillation Integrated with MDLM. Left: original MDLM architecture, masked positions are predicted based solely on the unmasked context. Right: Gumbel-conditioned architecture: distillation pairs $(\boldsymbol{\xi}, \boldsymbol{x})$ are obtained from an AR teacher; then MDLM is modified to take Gumbel as conditional input, learning to predict masked positions with corresponding Gumbel noise.

**Model Architectures and Baselines** We use GPT-2-Large as teacher model for MDLM and BD3-LM, both of which augments the diffusion transformer (Peebles & Xie, 2023) with rotary positional embeddings (Su et al., 2024). All model (including the AR baseline) implementation follows the transformer architecture from Arriola et al. (2025) that uses 12 layers, a hidden dimension of 768, and 12 attention heads, which corresponds to 110M parameters (excluding embeddings).

**Training** All models were trained for 1 million steps using the AdamW optimizer with a batch size of 512, and a constant learning rate of $3 \times 10^{-4}$ with a 2,500-step linear warmup. This translates to 65B tokens and 73 epochs on LM1B, 524B tokens and 60 epochs on OWT. For BD3-LM, we follow the baseline by training with maximum context length first for 850K gradient steps (effectively the same as MDLM) then fine-tune on the target block size for 150K gradient steps. We maintain consistent random seeds to ensure fair comparisons. All experiments are run on H100 GPUs.

**Nucleus Sampling** For all baselines, we follow their default setup and employ nucleus sampling (Holtzman et al., 2020) with $p = 0.9$. As discussed in many prior works (von Rütte et al., 2025; Wang et al., 2025), one could get suspiciously low generative perplexity values as $p$ becomes very small, at the cost of sacrificing diversity/entropy. We noticed that for models trained with Gumbel Distillation, the diversity of the samples is more robust against low $p$ values than the baseline, which could be attributed to the stochasticity brought by the randomly drawn Gumbel as the conditional input. To keep consistent with the baselines, we still use $p = 0.9$ for Gumbel-conditioned models.

**Gumbel-based Categorical Sampling for Diffusion Models** Zheng et al. (2024) analyzed the numerical issues of Gumbel-based categorical sampling for diffusion models and offered two methods for improvement, including (1) sampling 64-bit Gumbel variables instead of 32-bit to reduce the truncation effect, and (2) using their proposed first-hitting sampler. The BD3-LM baseline adopts both methods, which we choose to follow in our experiments.

However, it is worth noting that (1) Zheng et al. (2024) points out the token-by-token decoding process of masked diffusion models by their first-hitting sampler does not suffer from notable numerical

issues under 32-bit precision, which makes it unnecessary to adopt both methods at the same time, and (2) the first-hitting sampler effectively limits the masked diffusion model to unmask one token at each step, making its inference cost the same as AR models. Therefore, when we obtain samples using fewer NFEs than the generated sequence length, first-hitting is disabled and we fall back to the efficient ancestral sampler from Sahoo et al. (2024).

**Inference with Calibrated Gumbel Noise**   During training, the Gumbel noise $\xi$ is drawn from $\mathcal{G}(0,1)$ to fully capture the teacher's sampling variance. At inference, however, we observe that the heavy tails of Gumbel can occasionally produce rare, high-magnitude values that can lead to lower-quality samples. To ensure more stable and reliable generation, we can draw from a calibrated Gumbel distribution by introducing a temperature parameter $\tau \in (0,1]$ and scaling the guidance as $\xi \rightarrow \xi * \tau$ from the same Gumbel family but with a reduced variance, effectively controlling the influence of extreme values. Empirically, a slightly reduced temperature is found to bring substantial improvement in generative perplexity with minor decline in entropy, indicating that guidance from a lower-variance distribution is more consistently reliable. In our experiments, we use $\tau = 0.85$.

## C.2   MULTI-TOKEN-PREDICTION

**Medusa**   Prior to Medusa (Cai et al., 2024), various speculative decoding strategies (Leviathan et al., 2023; Chen et al., 2023) have been proposed to reduce the number of decoding steps for LLMs, where a smaller draft model is used to generate a token sequence, which is then refined by the original larger model for acceptable continuation. Instead, Medusa proposes using multiple decoding heads on top of the backbone model to expedite inference. These are additional decoding heads appended to the last hidden states of the original model. Specifically, given the original models last hidden states $h_t$ at position $t$, $K$ decoding heads are added to $h_t$. The $k$-th head is used to predict the token in the $(t+k+1)$-th position (the original language model head is used to predict the $(t+1)$-th position). A single layer of feed-forward network with a residual connection is used for each head. Denote the prediction of the k-th head as $p_t^{(k)}$, then the definition of the $k$-th head is:

$$p_t^{(k)} = \text{softmax}\left(W_2^{(k)} \cdot \left(\text{SiLU}(W_1^{(k)} \cdot h_t) + h_t\right)\right), \text{where } W_2^{(k)} \in \mathbb{R}^{d \times V}, W_1^{(k)} \in \mathbb{R}^{d \times d}. \quad (11)$$

$d$ is the output dimension of the LLMs last hidden layer and $V$ is the vocabulary size. $W_2^{(k)}$ is initialized identically to the original language model head, and $W_1^{(k)}$ to zero, which aligns the initial prediction of the heads with the original model. The heads are trained in conjunction with the original backbone model, which can be frozen (Medusa-1) or trained together (Medusa-2). At inference, a tree-structured attention mechanism is employed to process multiple candidates from each head concurrently. Empirically, Medusa found that five heads are sufficient at most.

**Model Architecture**   For simplicity and consistency with our other experiments, we choose GPT-2-Small as the backbone and follow Medusa-1's setup to freeze the backbone model during training. Since the open-source code of Medusa mainly supports certain LLMs, we re-implemented the core framework for our controlled-scale experiments on GPT-2-Small. As shown in Figure 6, we select the number of MTP heads as 4. Since each MTP head is designed to be a single linear layer, we first normalize and project Gumbel vectors via the same process as described for MDLMs, then we accumulate the Gumbel sequence via a small causal transformer and feed its output to the corresponding hidden state slice from the backbone as a condition for each head.

**Training and Inference**   We simulate a self-distillation setup, adopting the assumption that Open-WebText (Gokaslan & Cohen, 2019) matches GPT-2's output distribution. The MTP prediction heads are trained with a batch size of 256 for 20K steps as we observe further training does not improve the performance as much. At inference, we follow Medusa-1's setup and adopt typical acceptance as the strategy for speculative decoding, where the proposal is verified against both a hard probability threshold and a soft entropy-dependent threshold. Specifically, given $x_1, \ldots, x_n$ as context, when evaluating the candidate sequence $x_{n+1}, \ldots, x_{n+k}$, a candidate is accepted when:

$$p_{original}(x_{n+k} \mid x_1, x_2, \ldots x_{n+k-1}) > \min(\epsilon, \delta \exp(-H(p_{original}(\cdot \mid x_1, x_2, \ldots x_{n+k-1}))) \quad (12)$$

We select $\epsilon = 0.1$ and $\delta = 1.0$ for our experiments. To evaluate the acceptance rates, we adopt a simple sequential strategy and verify each candidate only when all previous candidates have been

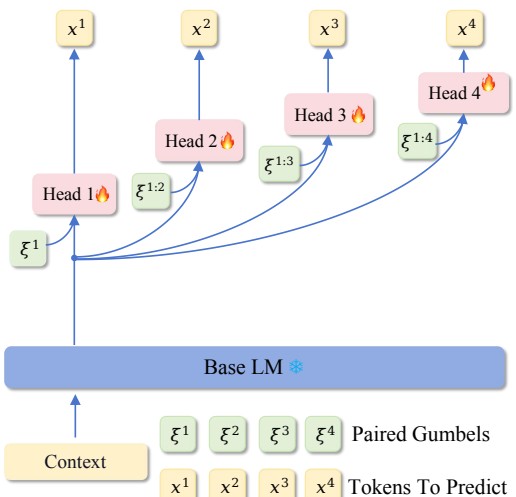

Figure 6: Gumbel Distillation integrated with Medusa. The base backbone model is frozen when training the MTP heads. We first obtain distillation pairs $(\xi, x)$ from the AR backbone as teacher, then each MTP head is modified to take Gumbel as conditional input.

accepted. We randomly select beginning slices of samples from the validation set of OWT and let the MTP heads complete after the starting prompt. Reported results are averaged over 500 samples.

## D  EVALUATION DETAILS

### D.1  ON EVALUATING GENERATIVE QUALITY OF DIFFUSION LANGUAGE MODELS

To quantitatively assess models' performance, we adopt standard language modeling metrics. For unconditional text generation, our primary metric is **Generative Perplexity (Gen. PPL)**, evaluated using a pre-trained GPT-2-Large model. Since GPT2-Large uses a context size of 1024, we follow our baselines and compute Gen. PPL for samples longer than 1024 tokens using a sliding window with a stride length of 512. The numbers reported are an average of 300 generated samples.

In addition to Gen. PPL, we also follow recent works (Wang et al., 2025; Fathi et al., 2025; Shing & Akiba, 2025) to report the **MAUVE score** (Liu et al., 2021) . While Gen PPL is a commonly used metric for measuring fluency, it can be an unreliable indicator of overall quality as it often fails to capture crucial aspects like diversity and creativity. A model can achieve lower perplexity score by generating repetitive or overly generic text that is highly probable under the reference model, yet this does not correlate well with human judgments of quality. In comparison, MAUVE score provides a more comprehensive assessment by comparing the distribution of model-generated text against the distribution of human text, offering a balance between generation quality and diversity that cannot be "hacked" by tuning sampling hyperparameters. Following Pillutla et al. (2021), we compute MAUVE score with 5000 samples from the validation set of OWT as the reference data.

### D.2  ZERO-SHOT BENCHMARKS

For downstream performance evaluation, we use the `lm-eval-harness` (Gao et al., 2024) library which allows custom models to run on benchmarks by providing likelihood estimation and text generation APIs. For likelihood-based multiple-choice tasks, we compute per-token likelihood over both context and completion (excluding padding). We adopt one-step evaluation, as this tests pure generative capacity without partial information, providing deterministic low-variance estimates that give more stable and discriminative results. The models perform one forward pass on the input where only answer completion tokens are masked, and the likelihood for each token is extracted from the model's predictions at the masked positions. For BD3-LM, block boundaries are respected by sequentially masking entire aligned blocks. Sequences exceeding model length are right-truncated to maintain recent context, similar to autoregressive context scrolling.

## D.3 LLM-AS-JUDGE

```
You are an expert text quality evaluator.  Please evaluate the given text sample from
these dimensions using a 1-10 scale with decimal precision.
  1. Clarity:  Assess how clear and logical the text is; Consider:
      - Is the main message/topic clear?
      - Do ideas flow logically from one to the next?
      - Is the text easy to follow and comprehend?
      Note:  Text may be an excerpt, don't penalize incomplete context.
  2. Grammaticality:  Evaluate grammatical correctness; Consider:
      - Proper sentence structure and syntax
      - Correct use of punctuation, capitalization
      - Subject-verb agreement, tense consistency
      - Absence of grammatical errors
  3. Factuality:  Assess accuracy of verifiable information; Consider:
      - Are stated facts (dates, names, places, numbers) accurate?
      - Does information align with common knowledge?
      - If no verifiable facts present, score based on plausibility.
      - Focus on internal consistency rather than perfect accuracy
  4. Style:  Evaluate the quality of writing style and fluency; Consdier:
      - Sentence variety and rhythm
      - Appropriate vocabulary and word choice
      - Natural flow and readability
      - Professional/appropriate tone for the content type
  5. Creativity :  Assess creative and engaging qualities; Consider:
      - Original ideas, unique perspectives, or interesting approaches
      - Engaging language and expression
      - Creative use of language, metaphors, or descriptions
      - Avoidance of overly generic content
The text to be graded is as follows:

"""
{text}
"""
```

Figure 7: Prompt used for LLM-based evaluation using Gemini-2.5-pro.

Following previous work (von Rütte et al., 2025), we evaluate the quality of the unconditionally generated text samples by using LLM as a judge. Specifically, we employ the Gemini 2.5 pro API (Comanici et al., 2025) to rate the samples based on clarity, diversity and fluency on a scale of 1 to 10. We provide the sample text along with instructions for the LLM to first give a justification and then grade each dimension accordingly. The final output from the LLM is restricted to a json format for parsing. See Figure 7 for the prompt template we used to instruct the LLM for the evaluation.

# E ADDITIONAL RESULTS

## E.1 MAZE NAVIGATION TOY

Here we present more details on the maze navigation toy's setup and results.

As shown in Figure 2, we used a $5 \times 4$ map for the maze. Denote the coordinate of the upper left point as $(0,0)$, our start point is set to $(3,1)$ and target point is set to $(3,4)$. Given a vocabulary of, {<bos>, <eos>, up, down, left, right}, we define a simple language as a sequence of actions that form a valid path from start to target in the maze, and set the maximum path length is to 10. We used BFS to generate 2000 valid paths as the training dataset.

For the AR model, its architecture is a transformer with 8 layers, 8 heads and a dimension of 128. We trained it with a batch size of 256 and learning rate of 3e-4 for 200 epochs to obtain an "ideal" teacher that almost always generates valid paths. Then, we implemented a simplified version of MDLM as student and trained it with/without Gumbel Distillation using the AR model as teacher. The student model is a transformer with 4 layers, 4 heads and a dimension of 128, and was also trained with a batch size of 256 and learning rate of 3e-4 for 200 epochs. At inference, we generate 100 paths using different number of steps and count the number of successful generations. For AR, NFE is fixed at 10 whereas we try NFE=1,2,4,8 and 16 for MDLM. The success rates are below:

Table 7: Success rates vs. NFEs for MDLM and MDLM + Gumbel Distillation on the maze toy.

| NFE | =1 | =2 | =4 | =8 | =16 |
|---|---|---|---|---|---|
| MDLM | 53.0 | 64.0 | 69.0 | 81.0 | 94.0 |
| MDLM + Gumbel Distillation | **82.0** | **86.0** | **97.0** | **99.0** | **98.0** |

## E.2 Likelihood Estimation

In Table 8, we also report the zero-shot validation perplexities. We evaluate the likelihood of models trained with OWT on Penn Tree Bank (PTB; (Marcus et al., 1993), WikiText (Merity et al., 2016), LM1B, Lambada (Paperno et al., 2016), AG News (Zhang et al., 2015) and Scientific Papers (Pubmed and Arxiv subsets) (Cohan et al., 2018). Since the zeroshot datasets have different conventions for sequence segmentation, we wrap sequences to 1024 and do not add eos tokens.

Note that the results here are *heavily* biased towards models trained with Gumbel Distillation as they are conditioned on the Gumbel vectors corresponding to the ground truth sequences. Since the model has learned a mapping from Gumbel noises to coherent AR outputs, it can generate good text samples using random Gumbel at inference, but this does not guarantee the prediction of the exact sequence on the validation set. However, we believe these results demonstrate how well the model has learned to estimate the conditional likelihood when the exact corresponding Gumbel is given.

Table 8: Validation perplexities ($\downarrow$) of models trained on OWT.

| | PTB | Wikitext | LM1B | Lambada | AG News | Pubmed | Arxiv |
|---|---|---|---|---|---|---|---|
| MDLM | 92.54 | 34.52 | 66.10 | 48.63 | 65.32 | 43.67 | 37.47 |
| MDLM + Gumbel Distillation | 35.12 | 15.57 | 26.01 | 15.56 | 20.03 | 20.81 | 16.85 |
| Block Diffusion | 96.27 | 30.82 | 61.14 | 49.51 | 63.09 | 42.59 | 39.39 |
| Block Diffusion + Gumbel Distillation | 37.54 | 13.86 | 22.69 | 15.78 | 18.19 | 19.78 | 17.02 |

## E.3 Infilling Evaluation

We evaluate whether Gumbel conditioning preserves the versatility of masked diffusion models beyond unconditional generation. Following the setup in our rebuttal, we sample sequences of length 128 from the LM1B validation set, uniformly mask 32 tokens, and use the same MDLM sampler as in the main experiments to infill the masked positions. We report GenPPL computed *only on the masked tokens* using the same AR evaluator as in Table 1, averaged over 300 samples.

Table 9: **Infilling on LM1B.** Gumbel conditioning improves GenPPL on a non-AR infilling task, suggesting it does not overfit to AR-style sampling.

| Model | GenPPL$\downarrow$ |
|---|---|
| MDLM | 55.1 |
| MDLM + Gumbel Distillation | **47.8** |

## E.4 Model-Size Scaling

We also study how Gumbel Distillation behaves when scaling MDLM. We compare a small model (12 layers, $H$=768) and a medium model (24 layers, $H$=1024), where the medium models are only partially trained (40% of tokens seen compared to small models), and the trend persists.

Table 10: Gumbel Distillation improves both MAUVE and GenPPL for different model scales.

| Method | MAUVE$\uparrow$ | GenPPL$\downarrow$ |
|---|---|---|
| MDLM (Small) | 0.179 | 78.74 |
| MDLM (Small) + Gumbel Distillation | **0.264** | **67.64** |
| MDLM (Medium, partial) | 0.186 | 74.22 |
| MDLM (Medium, partial) + Gumbel Distillation | **0.253** | **61.34** |

## F Generation Samples

'The two Houses have adopted a bipartisan approach, inviting new rules and licensing releases for federal agencies. Currently, they require the agencies to have in-depth internal review of the rules. They agreed on rules and didn't stop until Banking and Budget Chairmen shared the Senate Finance Act through Senate Finance Bill 2012, though there is no immediate support for any reform. Democrats are not united in wanting a near-complete return to the rules, which would require months of political pressure to rein in regulators.While some Republicans now have pre-existing differences, Majority Chairman Jim Cantor (R-West Virginia) has softened his stance. He said the House will continue support committee work on a bill to help break the rules but also create a complex overhaul.But Mr. Cantor said that, I sort of implicitly trust that we will pass the bills in a bipartisan way with unified Republican leadership. Still, Democrats continue to be cautious about what it would cost for the 2012 financial bill to pass. Mr. Cantors measure is to be the final one that cleared the Senate next week. The House will return it on the floor for consideration if it passes rules, Mr. Dehner promised.Senators and Republicans say they never expressed support for much more oversight of federal regulators because of the controversial Dodd-Frank overhaul in late 2013. Democrats had resisted plans for several other Dodd-Frank initiatives to go beyond the beltway and create a line for federal oversight. They feared that financial regulators would set up a revolving door and endless financial-related regulations could slow data collections and satisfy people.But Senator Daniel D. Coats (R-Ohio) offered flat tax increases to create the height-line approach. Now, the idea is are regulatory overhauls, Financial Services Chairman John Thune said Additional reporting by Benjamin Reeves<|endoftext|>By Edward Snowden: Susan Onghi Kara CORRECTION: John Toner says ""Uhoh pills"" on record. Forty to 2, Edward Snowden. SiriusXM Broadcast A few days ago, Frank Geithner spoke on The Friends Hour of 11 News Bloomberg about defending the ground line in the fight against Pragmatism and saying that the only alternative in the matter requires a reading of the Elements Policy of Medics.Neither, though, should draw a line on important and essential issues.He said that an alternative is necessaryafter an important decision stage for Russia. Vitaliy Goldifkin earlier had outlined a set of decisions with major implications. Russias actions will risk a layer of sanctions and the ire of many Republicans in Congress.He warned that regional actors are in conflict on a wide level, saying that the West perceives that power and influence grandly come to dictate strategic posture, economic, military, and political determination. He passed on saying except tranquility with regards to the nuclear weapons regime, there could be a result of direct, direct disruption of the Russian economy. But after just enough debate and more explanation, it had emerged: The treaty must renew and be upheld. Ultimately, the United States must rely upon our allies to sift through the facts. Russia made the nuclear commitment publicly, on the possibility of a possible joint military strike. Nevertheless, any type of action, through the physical and through the enforcement of the treaty, is a priority of the United States, Rouhani said. The European Union have already indicated they would punish Russia with a retaliatory measure, Russia would impose other sanctions, and far China China look to develop an immediate utter surprise of retaliation, considering Chinas economic policy, which applies if Putin seeks it. Putin talked like way about a potential reshuffle of Saudi Arabia and Iran. The Saudis themselves could just show their belligerence toward Russia or encourage the United States to use force to quickly squeeze Russia. But Obama has given the Saudis an excuse to berate them diplomatically for slippage, while relish to suggest that the United States is consummate engaged in sanctions against Russia. That situation continues with the Russian response. The report that Russia is about to install a nuclear missile shield on U.S. One waves the Twitter prompting of unnamed Russian statesman Andrei Lavrov. Others chimed ramefully by sounding the following bell. Its not gained from you. Author Vashemi Kopacak stepped in for Putins explanation. That disregards what you sent over cross borders with your hands, what you chose to eat and listen to strangers while people at or having intel leak out to your enemies. Why they cant you put your weapons to the enemy and calculate it.No'

Figure 8: Sample from MDLM of length 1024 and T = 1024 steps. The generative perplexity of the sample under GPT-2-Large is 62.36 and its entropy is 5.56.

```
 'The Japan Galactic Institute has announced that it will become the world's ever-growing space lab.  The
company will be developing a commercial system and used to move Japan's space ship into the next frontier.
The market is booming and the central Japanese market forecasts revenues of $2.6 billion to $3.8 billion
in 2013.""Japan Galactic is a big step forward in getting used to in Japan, which has the ability to be
the biggest of the world,"" said Jean Phillips, chief executive of European Science Systems (ASES), an
international research company.  ""As for the Japanese government's success in foreign policy, we're proud
that the CNAS is a partner.  We look like the Japan Galactic Institute is a huge step to something of
a world-type science laboratory.""Kashisei and its partners have been developing Japan as a global hub
for industries such as solar power and tourism.  Starting in the early 1990s, Japan saw building towns,
cities, villages, towns, rural areas and villages a quarter to a quarter and a third of the world's GDP.
In 2010, the Nordic country recorded an annual GDP of $1.11 billion and grew by $14.1 billion faster than
originally projected.  Japan's global GDP grew to $14.8 billion, $181 million in 2013, and $130 million,
with Asian nations slowly adopting new varieties of seed for crops and indigenous seeds for their own
growing economies and economies.That's because Japan has managed to use the technology to get a website
for online courses and a new marketing technology, too.  As is being developed at Japan Galactic, the
CNAS will be able to use a software platform, directly from sensors, chips and computer equipment, to
connect the lab to the internet, so anyone can have to connect the technology to a network in order to
prove themselves.The work behind the science study will have to be a bit harder.  Once Origin Incorporated,
the in vitro technology, the cornerstone of Japanese research, will be patented.Until this point in time,
the company will have been seeking to fly to and around the European Union.  Now it is soon moving a small
European research firm into the institute's systems and hardware to make it the largest possible market in
the next few years.""We want to aim to take the lead in a scientific lab that extends its capabilities all
over the world.  We want to see competitive research in pursuit of our own lives.  And if this happens,
it's a huge breakthrough in the human race and it's a very exciting opportunity for innovation,"" said
James Smith, the head of the European Union's main scientific research organization, in his remarks
last Wednesday.  First, the World Congress will launch a similar technology in July and SpaceShipOne's
first public flight in June.  But it's the first major launch ever in the US. One first year to take US
experiments together was announced by a consortium that had made headlines.  Although the European market
is already the first to use the technology, not all companies are doing the same.SARC  a medical group
whose works show the potential to provide the treatment for cancer  is moving to use the European Union,
government, agencies and government scientists to set up its own research laboratory on CNAS. This is a
trip up to space, and both experiments will carry a drug and brain tested.The first stage is that it lets
the US take a part in its orbit.  Unlike the first spaceflight process, it must be transited and overseen
by a plane.  The technology also lets the US to interact with the environment, including in places like the
International Space Station.  That probably leads to its own bigger push into the robotic body.  But now
a second stage is where the FDA is looking to find ways to make comparisons and communicate to the public
about how to deliver drugs.  It's a way that the effects on the human body have been far less measured.CNAS
still needs approval of its patent.  Using the technology that could produce therapeutic efficacy in those
with psychiatric disorders  a broad vein of neuroscience, medicine and data  would violate several of
the agency's guidelines.""We seem to find that a few years progress on the case would be slow and hard
to justify because it contradicts these government and the need for higher quality,"" Smith wrote in his
remarks.The U.S. Senate in July has suggested it will not approve any proposals that boost sales.A similar
measure in May led Europeans to vote on whether or not drugs should face peer-to-peer trading.The news
dark off signs in 2015 that the European Parliament passed the European Commission's next financial year,
inhibiting the FDA's scientific review of its potential effect on certain drugs.A pilot for a similar study
is expected to cost about $2,000 each and is expected to launch in the UK, the end of 2016.<|endoftext|>The
Bank of England announced the interest rates on December 7, recommending a discount rate of from 28
```

Figure 9: Sample from MDLM + Gumbel Distillation of length 1024 and T = 1024 steps.  The generative perplexity of the sample under GPT-2-Large is 35.19 and its entropy is 5.32.

```
 'Two years then signed him, and he told his friends.  He was one of the best soccer players ever.  helped
take out the England national team, which is vacant at this stage.  It's been a long journey, said
Rodriguez, who's now gone beyond the borders of the world.""I'm going to say I love England,"" but to be
said England, and to have fun.""Obviously he's very close.  I have no doubt he is a highly respected male
partner of my country,"" said Rodriguez, who signed as ambassador.  ""But not wanting that much to beat
the player, because, like a passion.  He's going to be here for a long time just to go there."" Rodriguez,
like the England Americans and others, has been a huge player to get a top-flight team in the top six
would be tough.""He's been on the right side,"" Rodriguez said.  He's England sooner than I was.  The last
time I was with the European team.  He's been really tough, whether we get to the fourth round.  We have
been competitive, the capital of the Southwest region.""Demophia Deillen, who was a first-round pick, one
of the best players in the world, who has played heavily in hopes of winning the national championship
in the past two years, even as part of a loan by the United States to Colorado part.""But he's not an
idol.  We don't have that many other guys in the soccer world with the highest-level play -- Chicago but he
knows his side,"" Deillen said.She said in a statement after England's World Cup title:  ""He has helped
a team that signed a numbers-like season in the National Soccer League.  That's exciting football.""""""I
care"" women's soccer players more at Manchester City than anywhere in the European League of ""everyone
is in the United States,"" International Director Brian K. Menjelstra told The JTA.  ""I am happy to win
World Cup and to have a person able to represent the women's league in England and with women's Soccer
at the top of my heart and inspire me to defend my country.""The United Soccer Association currently
selected women's first team competition winners, and wishes competition organizers to prepare the best
young players beyond the age of 30.Meanwhile, the academy is in Australia and begins in New Zealand.  It's
now the biggest and biggest tournament the women team in America has played.""It's a challenge for the
fans see women's Soccer playing in the World Cup finals,"" Deillen said.That the England Americans had the
players they signed.  A lot of them have seemed to say that they can't stand these clubs, and this time
they are players themselves.""That would be a concern.  I think you look at some teams, one year and-16,
some of them hoping to get them in four years,"" she said.  So far, the England players appear to have
no nationalistic mind.In the preliminary stage, there are no signs that the two national players left the
team.  They have no say on who is now, and the team is still with the England players.  I know there's yet
another team to play for me and that's really my confidence.""Nevada spokeswoman Laura White later issued a
statement to the federation declaring this ""a new one"" between the two sides, which will play in London,
as for the kind of spot where the top two are in contention by the two federations.White spoke with the
federation at 9:40 p.m., and they said the players didn't be ready to take part before, a statement that
indicates the way the women's federation intends to be used.White said the morning before the tournament
said he could not be with that team, and that he was prepared to make himself unavailable.  He has been
contacted by the United States Embassy for comment on the price he had paid.Shake said she is ""where the
players are for a while,"" but added that she will continue to be connected to the team.<|endoftext|>Mike
Carrington (SMS chief) made his case to Clarke Hall (SSP), who heads Royal North-C. Wales for the west
west Wales University Hospital, saying the two are ""very close associates on the NHS"" as a result of a
phone call.""My first reaction is I'm not happy at the moment,"" said Royal's deputy chief, a nurse who was
traveling last week in his clinic to meet Hall and hospital.On Tuesday morning, a spokesman for the chief
""called into similar situation with our partners, who were working with the NHS, where they approached
our officers at the same conversation with the spokesman,"" he said.North-west Wales chief director of the
hospital posted official website, saying:  ""If there are no good answered on the NHS's mobile phone then
```

Figure 10: Sample from BD3-LM of length 1024 and T = 1024 steps. The generative perplexity of the sample under GPT-2-Large is 37.7 and its entropy is 5.21.

```
'How many times in my life have I been a woman?  I've always felt like a woman in everything I've done to
be proud of, just being a great person, so I am reconsidering.  My gifter wants to be named the ""Santa""
and we will have to see her on Christmas Day.I have received many gifts over the years, but I am a soppin'
and bemo in my body, I feel like I am a woman!  All I want to throw away this thing is to do a happy
and productive job like that.Something has been happening a lot lately, and it's a very frustrating time
today.The gifting is just part of a series of events in the past year.  The location is in Orlando, and
a relatively long way away from home.  The photos are the ones I want to give to the community, not the
ones my senders are not getting.  But I know the ones I'm sending are donating to or assisting the fans
in these gifs.My gifter will be thanking my Santa and giving me hugs!  If I are you, you are in my heart.
You are here at my funeral.  This is what I am all doing.  I don't feel like being a hero.  I'm literally
not giving a damn about the people that I put to them.  It's beautiful, but the picture isn't about my
heart?  If you are me, it would be more than what I do.And it's not something that a lot of people do.  If
I'm shown in a video, I just see some of the news pieces that have been posted and on.  I want for people
to show their support for (my gifter) and they will tell me.  I'm not doing what I do.I have to give away
the gift, I'm not a ""artist"" or an artist at all, and I'm not a ""real creator"" of anything to leave
my own name, and I don't want to be a woman to get any support.  I want to get a name on the internet
for this thing.  You aren't getting the name out of people who deserve to be given a name.With that said,
at least one of the people in this individual's circle have been kind through me.  I have never thought
while doing my rounds with the gifter I would give me away.  I just laughed.  I thought gift exchanges
would make me feel better.  Or maybe they would have.  But I'm not leaving it up to the gifter on it.
It's just acceptance.  A friend was my mother, but he said that she likes helping people, and she said he
wants to get away from things that have been done.  However, she didn't make any plan to gift him.  He was
completely aware of it and just thought and went with it.  I want things like this to get away from all
of my gifs.  I think I have only one thing to do.  Both of those things are bad and I cannot gain access
to them.  These are bad things and when I read about them, I find them justified.  I then give the gift
to gifters and I take it to the audience to show me that I can do what I could do.  I want to be able to
get something to give to them, and give it to others to give away.  I'm just then going and then tapping
into the pressure on me that is stopping them from helping me.This isn't simply a gift.  When I've done it,
they are usually getting it away anyway from all of them.  I don't know how all I did, I wasn't thinking
about it, but I went on my way.  I put up this opportunity for people that I was going to consider and
have a very happy time.  It's basically all about me.  I was going to give it on to someone who was one of
the ones I was after and I didn't even have to give them the support because I was going.  It didn't have
felt right for me, as I went to what I already did and I just never got the tool that I had.  I'm so happy
with it.  I spent a couple of months getting to get that power back and go back to the world to write about
myself.  I have not really defined myself as a real artist for being a.  So that is a good feeling for all
me.  The people do not want me to get through the process of loving as an artist, not wanting to be me to
be the person I can be.  I need to reach out to my fans, to know what the people have against me.  But I go
through this.  I need to show my support for them and gifting.  The fact that those are all benefits the
people have, of the community itself.  That means I have a lot of tiny self-help projects and projects that
are awesome.'
```

Figure 11: Sample from BD3-LM + Gumbel Distillation of length 1024 and T = 1024 steps. The generative perplexity of the sample under GPT-2-Large is 27.2 and its entropy is 5.19.

```
 'The numbers game to me, the most important takeaway is the media wants to sensationalize the story, it
wont matter.  The real story here is what I have come to expect of the average reporter.  I have come to
expect that most things you say on the air will blow up and overshadow the fact that you are a regular
citizen and could well have said anything you want to people if they catch on.  But this isnt the case.
The average anchor on CNN has changed the story on a regular basis.  If you dont listen to the media,
you will lose.In the name of being unbiased, the mainstream media will remain a relatively unknown group.
I think they are having an effect on the perception of the public.  In my view, everyone on the air is
over-represented on network television.  If they are not correct, they can win over viewers who may not
necessarily be paying attention.  A good example of this would be when CNN featured a big story about the
North Korea nuclear test and the ratings went down.  For some reason, the coverage led to fewer viewers
on CNN, and that was definitely what happened.I think the U.S. media is more adept at making propaganda
pieces.  The result is that when they present a big story, and when they get the impression they are
going to win, people might not go on CNN. I expect that my expectation is that the news media will get
more viewers on CNN and possibly even win more viewers for cable networks (not to mention newscasts and
prime time shows).  But if they dont, I think a new wave of viewers might be swayed to the second tier.\n
\nThe outcome for me is unclear.  CNN has a big story, and I do expect it to draw in a large percentage
of viewers, while the anchors are losing viewers.  This could be an entertaining topic that the major
networks want to have on air.  Maybe they want to win over viewers for those networks?  The next step I
see is for them to stop showing old news if they dont think the American public is paying attention.You
can follow @Nero on Twitter.Commentscomments<|endoftext|>Disneyland will have you believe the first time
you step inside The Collectors Mini Tower on the grounds of Disneylands Hollywood Studios.  As the world
and park matures, people are returning to visit Disneyland for the first time.Construction on the 10,000
square foot star-studded attraction began in 2001.  Disney developed the idea to include the area as
a Disneyland Memorial Hall in the park.In August 2013, The Collectors Mini Tower was revealed to the
public during a visit.  During the visit, guests saw the incredible view that was included inside the
tower and learned about how much it cost to construct this unique attraction.According to Disney, the
five-storey building is 42,000 square feet and features more than 1,500 rooms, with windows overlooking
the street level of the exterior to the inside.The size of the structure makes it a unique experience
for tourists.  The attraction itself takes guests to a special foyer lined with board game and ride
setups.The Tower was constructed at a cost of \$60 million and includes several areas of state owned
amusement park property.<|endoftext|>Both the Canada Goose and the Canadian flag may soon have new
colours.The federal government is changing its policy that says any bill to change Canada's flag must
be approved by parliament, making the Canada-U.S. trilateral trade agreement less of a reality.\n \nLast
month, Finance Minister Bill Morneau issued a memorandum to all government ministers outlining what he said
would be changes to the copyright and trademarks laws if the Canada-U.S. trade agreement between the two
countries is not approved by Parliament.Under the memorandum, the government is proposing to replace the
country's Canadian half with an intergovernmental and unionized version of Canada's flag, or the flag of
the Confederation.The new agreement between Canada and the U.S. will add a new label to the Canadian flag
called the C Treaty.  It marks a change from the government's Conservative government of prime minister
Stephen Harper's to a Liberal government of prime minister Justin Trudeau.It is estimated that 12 million
Canadians could lose their current U.S.-ca.  Canadian half.  It is almost impossible to pick a colour
without buying one.  – Patrick Simon, retired Ottawa journalist.To get your product into the C Treaty, the
country must submit its new government agreement, called the C Treaty, to the U.S. Congress and ask the
administration to pass it.When the C Treaty is approved, it will be trade liberalization, which will start
with a six-month notice period from October 1, 2018, through October 31, 2019.  That means the new treaty
will be in effect for three years.According to the Finance Minister'
```

Figure 12: Sample from AR of length 1024 and T = 1024 steps. The generative perplexity of the sample under GPT-2-Large is 15.7 and its entropy is 5.29.

