# OpenReview forum: "Gumbel Distillation for Parallel Text Generation"
_ICLR.cc/2026/Conference — ICLR 2026 Poster_

### Official Review · Reviewer_B2R6 · 2025-10-24

**Soundness:** 2
**Presentation:** 3
**Contribution:** 3
**Rating:** 4
**Confidence:** 4

**Summary:**

The paper introduces a method called Gumbel Distillation that turns the teacher model’s sequential sampling process into a deterministic "blueprint" of noise values which a parallel student model can condition on, enabling high-quality parallel decoding.

The authors show how to extract these noise "blueprint" from teacher samples or recover them from existing text using a parallel posterior sampling procedure, and they explain how to inject the resulting signal into masked or multi-head parallel decoders so the student learns to reproduce the teacher’s joint decisions.

Empirically, this approach improves generation quality compared to prior parallel decoding methods across multiple datasets and model architectures, with gains on metrics that capture both distributional similarity and sample fluency, and with ablations that demonstrate the importance of the Gumbel formulation and the posterior extraction step.

The paper also provides practical algorithmic details and theoretical justification for the posterior extraction, and it discusses limitations and directions for scaling the technique to larger vocabularies and more structured noise representations.

**Strengths:**

1. The paper raises a good question: improving the accuracy (MTP) and quality (DLM) of parallel generation is a meaningful endeavor that will help improve the inference speed of autoregressive models.
2. The paper is clearly written and very detailed, presenting comprehensive methods, experimental details, and theoretical proofs, along with specific case studies.
3. The paper proposes a simple, commonly used, yet effective method that improves generation quality in DLM and increases acceptance rate during MTP.

**Weaknesses:**

1. I think the experiments are still not solid enough. You should compare them with other existing methods, such as other distillation methods. While it's good to see improvements when adding a new method to the original model, you need to compare it with similar methods to see if it outperforms them.
2. I'm not sure how the green gains in Table 3 are calculated. The difference between the two doesn't equal the green value. I'd also like to know the average generation length, as the current improvement in acceptance rate is minimal. It would be even better if you could provide the variance over 3-5 runs.
3. I think MTP has greater potential, as few people would retrain pre-training and DLM just to add a new method. Therefore, I think you could conduct several similar experiments with related methods to make your results more convincing.

**Questions:**

1. The authors acknowledge in the Limitation that computational complexity increases with vocabulary size, but current models often use very large vocabulary. For example, the commonly used Qwen is nearly 152K. I'd like to know how many times more computational effort is theoretically required compared to GPT-2?
2. I'm not sure why Head 0 is also trained in Table 3. I understand that Head 0 is essentially the LM head, and Medusa doesn't modify it, otherwise it wouldn't be lossless. Is this because gumbels must be input continuously? Are there any experimental results demonstrating true losslessness?
3. Comparing Table 4 with Table 1, I find that Sequential Gumbel Extraction's MAUVE score is better than the baseline, while PPL is worse. It seems that the introduction of gumbels only improves the Parallel model, but the two are theoretically equivalent. What are the specific implementation details of Parallel and Sequential? Does Sequential use a Casual Mask for multiple forward passes, while Parallel performs a single forward pass without a mask? If yes, are the Logits very different between the two? I feel more explanation is needed here.

---

> ### Author Response · Authors · 2025-11-23
> **Response to Concern 1**
>
> We thank the reviewer for highlighting the conceptual clarity and generality of our paper, as well as the constructive feedback. We will address each of the concerns below:
>
>
>
>
> **Response to Concern 1 (Lack of Comparison Baselines):**
>
>
> We agree that our paper could benefit from direct comparison with classical AR-supervised NAR distillation baselines as well as discussion on their differences. Following this suggestion, we have run new experiments to compare Gumbel Distillation against:
>
> - **Token-level distillation (“soft” KD)**: the student is trained to match the teacher’s full logit distribution at each position via a KL divergence term in addition to the standard loss, as in classical knowledge distillation [1].
>
> - **Sequence-level distillation (“hard” KD)**: the student is trained on sequences generated by the AR teacher (teacher’s generations are treated as ground-truth labels), following sequence-level knowledge distillation [2,3].
>
> - **APD (Adaptive Parallel Decoding)** (as suggested by Reviewer pK43), a recent, training-free method for accelerating diffusion LMs by using an extra, small AR verifier to adaptively accept the longest high-quality generated sequence[4].
>
> All methods are evaluated on MDLM trained on LM1B with the same backbone and evaluation protocol as in the main paper. The new results are as follows:
>
> | Method                           | MAUVE Score ↑ | Gen. PPL ↓ |
> |----------------------------------|---------------|------------|
> | MDLM                             | 0.179         | 78.74      |
> | MDLM + Token-level Distill       |      0.166         | 95.88      |
> | MDLM + Sequence-level Distill    | 0.169         | 99.48      |
> | MDLM + APD [5]                   |    0.203           |  57.61        |
> | MDLM + Gumbel Distillation       | 0.264         | 67.64      |
> | MDLM + Gumbel Distillation + APD |    0.255         |   49.28       |
>
> As the new results demonstrate:
>
> 1. **Token-level distillation** fails to address the core issue, only forcing the non-AR student to align with the AR teacher's per-token marginals, under the same conditional-independence assumption within a parallel block. In our experiments, it even **degrades** Gen. PPL relative to the baseline, showing that matching marginals alone is not sufficient for robust joint modeling and can even contradict the original learning objective.
>
> 2. **Sequence-level distillation**, where the student is trained on hard teacher-generated tokens, shows only minor to no gains in quality compared to Token-level distillation (MAUVE slightly below the baseline and much worse Gen. PPL). We hypothesize this is because teacher-generated corpora tends to be narrower, and the student can inherit teacher sampling artifacts and suffer from mode collapse.
>
>
> 3. **Gumbel Distillation**, in contrast, significantly improves both MAUVE and Gen. PPL over the original MDLM model and over both distillation baselines. Rather than distilling the outcomes (logits or generated text), it distills the **sampling process** itself: the student learns a deterministic mapping  $(x_{\neg I}, \xi_I) \mapsto x_I,$  where $\xi_I$ is the Gumbel “blueprint,” which serves as a shared latent that coordinates all tokens within a block and directly targets the joint dependencies that **classical methods do not address**.
>
> 4. **Adaptive Parallel Decoding (APD)** can be applied on top of both the baseline MDLM and our Gumbel-distilled MDLM without retraining the backbone. In our extended experiments, we observe that APD + Gumbel Distillation yields the best Gen PPL, as APD benefits from the stronger proposals learned by our training-time method. However, this comes at a modest cost of diversity, as evidenced by the slightly worse MAUVE score.
>
>
> **References**
>
> [1] Hinton, Geoffrey, Oriol Vinyals, and Jeff Dean. *Distilling the knowledge in a neural network.* arXiv preprint arXiv:1503.02531 (2015).
>
> [2] Kim, Yoon, and Alexander M. Rush. *Sequence-level knowledge distillation.* Proceedings of the 2016 Conference on Empirical Methods in Natural Language Processing (EMNLP), 2016.
>
> [3] Jiatao Gu, James Bradbury, Caiming Xiong, Victor O. K. Li, and Richard Socher. *Non-autoregressive neural machine translation.* arXiv preprint arXiv:1711.02281, 2017.
>
> [4] Jungo Kasai, Jiatao Gu, Marjan Ghazvininejad, Luke Zettlemoyer, and Mohit Iyyer. *A disentangled context approach to non-autoregressive neural machine translation.* International Conference on Learning Representations (ICLR), 2021.
>
> [5] Israel, Daniel, Guy Van den Broeck, and Aditya Grover. *Accelerating Diffusion LLMs via Adaptive Parallel Decoding.* arXiv preprint arXiv:2506.00413 (2025).

---

> ### Author Response · Authors · 2025-11-23
> **Response to Concern 2, 3 and Question 2**
>
> **Response to Concern 2 and Question 2: Clarification on the MTP Experimental Settings and Results**
>
> To clear up any confusion about the experimental settings in the MTP section, we want to make the following clarifications:
>
> The green numbers in Table 3 are **relative improvements**, not raw differences. For example, the 22.0% increase on Head 3 is computed as $\frac{0.322 - 0.264}{0.264} = 22.0\% $. In the revision, we will make the calculation explicit in the paragraph and change the notation to “Δrel%” to avoid confusion.
>
> For the MTP experiments, we randomly select beginning slices from the OpenWebText validation set, use them as prompts, and let the model generate a fixed 256-token continuation. At each decoding step, the heads all propose candidate tokens for their respective offsets; these are then verified sequentially by the base LM as in Medusa’s “typical acceptance” scheme. The “accepted length” we report is the *expected number of tokens accepted per decoding step*. Specifically, for the baseline, with acceptance rates $(a_0, a_1, a_2, a_3) = (0.987, 0.445, 0.310, 0.264)$, this is: $ a_0 + a_0 a_1 + a_0 a_1 a_2 + a_0 a_1 a_2 a_3 = 0.987 + 0.987 \cdot 0.445 + 0.987 \cdot 0.445 \cdot 0.310 + 0.987 \cdot 0.445 \cdot 0.310 \cdot 0.264 = 1.609 $.
>
> We will explain this formula more clearly in the main text to make the notion of “Accepted length” more transparent.
>
> We also would like to clarify our Medusa training recipe. There is **no restriction imposed by Gumbel inputs** that forces us to train Head 0; we chose a symmetric treatment for simplicity and **do not modify the backbone LM head** itself and keep it **frozen** during training.  The **auxiliary proposal layer at position 0** is introduced analogously to the auxiliary heads at positions 1–3, but the verification step can definitely continue to use the original LM head (as in Medusa), such that the base LM remains intact and can be used on its own.  To obtain strict “losslessness” , one can simply drop the auxiliary proposal at position 0 at inference time and use only the frozen LM head; our training procedure does not prevent this.
>
> Empirically, the Head 0 acceptance remains essentially unchanged (0.987 -> 0.990) after training, indicating that the NTP behavior is not degraded. We agree that auxiliary Head 0 is not strictly necessary and will clarify in the next revision that one can simply train heads beyond the next-token-prediction position while keeping the original LM head entirely untouched.
>
>
> **Response to Concern 3: More Experiments with Gumbel Distillation on MTP**
>
> We agree with the reviewer that, in practice, MTP can be highly attractive, since users are more likely to add Medusa heads on top of an existing LM than to re-train a diffusion LM from scratch. To strengthen this part of the paper, we conducted additional MTP experiments on a larger backbone from Medusa’s official codebase, namely Vicuna-7B, a LLaMA-based open-source model.
>
> We follow Medusa’s recommended training setup:
>
> - Base model: Vicuna-7B (frozen).
> - Data: 60k ShareGPT samples.
> - Baseline Medusa: reuse the base LM head for next-token prediction and train **3 additional Medusa heads** (Heads 1–3), keeping the base model weights frozen.
>
> We then apply Gumbel Distillation to the Medusa heads (again, without modifying the base LM head), and report per-head acceptance rates in the same format as in our paper:
>
>
> | Head | Baseline (Vicuna-7B + Medusa) | + Gumbel Distillation |
> |------|--------------------------------|------------------------|
> | 0    | 100.00                         | 100.00 (LM head)      |
> | 1    | 54.46                          | **59.33**             |
> | 2    | 30.25                          | **38.76**             |
> | 3    | 21.39                          | **29.42**             |
>
> These results show that our method continues to provide clear and consistent gains when applied to a larger backbone. In the next revision, we will include this table and a brief discussion, further supporting the reviewer’s point that Gumbel Distillation is particularly compelling in the MTP/Medusa setting.

---

> ### Author Response · Authors · 2025-11-23
> **Response to Question 1 and 3**
>
> **Response to Question 1: Limitation on Vocabulary Size**
>
> Similar to our response to Reviewer PK43, we agree that conditioning on a vector of size `vocab_size` raises natural questions about scalability. We already flag vocabulary size as a limitation in Section 6; here we clarify how the cost behaves and why we believe the issue is primarily about *optimization at extreme vocab sizes*, rather than an inherent barrier for large LMs.
>
> Our Gumbel injection adds a **single projection layer** that maps a Gumbel vector of dimension `V = vocab_size` to the hidden dimension `H`. This introduces a fixed parameter and compute cost of order
> $\mathcal{O}(V \times H)$, which is applied once per sequence, and does **not grow with model depth**. In contrast, the base transformer scales as $\mathcal{O}(L \times H^2)$, where \(L\) is the number of layers. As a result, the *relative* parameter overhead of the Gumbel pathway actually **decreases** as models scale up:
> $$\text{Gumbel Parameter Overhead} =
> \frac{\text{Gumbel Cost}}{\text{Gumbel Cost} + \text{Base Cost}} = \frac{V \times H}{V \times H + L \times H^2} = \frac{1}{1 + L \times H / V}. $$
>
> Using typical vocabulary and hidden sizes for several standard models, we obtain:
>
> | Model Scale   | Layers \(L\) | Hidden \(H\) | Vocab Size | Total Params | Gumbel Param Overhead |
> |---------------|--------------|--------------|--------------|------------------------|------------------------|
> | GPT2-Small   | 12           | 768          | 50257 | 117M         | 32.9%                  |
> | GPT2-Large   | 36           | 1,280        | 50257| 774M         | 8.3%                   |
> | LLaMA-7B      | 32           | 4,096        | 32000 | 7,000M       | 2.9%                   |
> | LLaMA-13B     | 40           | 5,120        | 32000 | 13,000M      | 2.0%                   |
> | Qwen-7B | 32 | 4096 |  151851| 7,000M | 8.1% |
>
> As the table above shows, the relative overhead is relatively high only for very small models; for LLaMA-scale models it drops to just a few percent. A similar reasoning applies to FLOPs: the extra softmax + projection is a tiny fraction of the total transformer compute once \(L\) and \(H\) are large. The only extra addition is the teacher model’s forward pass (once per sequence, at training only).
>
>
> **Response to Question 3: Clarification on Sequential vs. Parallel Extraction and Ablation Results**
>
> We appreciate the reviewer’s keen observation regarding Table 4. We clarify the specific implementation differences and explain why the results diverge despite the theoretical analysis.
>
>
> **1. Implementation Details**
>
>
> - **Sequential Extraction Uses Ancestral Sampling**: This method generates synthetic text from the teacher. We run the AR teacher step-by-step (using a standard causal mask) to perform ancestral sampling. At each step, we sample Gumbel noise $\xi$, and calculate $x_i=\arg⁡max⁡(l_i+\xi_i)$ (categorical sampling via Gumbel-Max trick), and feed $x_i$ as input for the next step. The result is that the student is trained on teacher-generated text $x_{synth}$ paired with the noise $\xi$ used to generate it.
>
> - **Parallel Extraction Uses Posterior Inference):** This method uses real, high-quality human text (e.g., LM1B/OWT). We feed the full human text sequence $x_{real}$ into the teacher in a single forward pass (still using a causal mask as in standard AR forward process). We then sample $\xi$ from the posterior distribution (Theorem 3.1). The result is that the student is trained on human text $x_{real}$ paired with the "inferred" noise $\xi$ that would have produced it.
>
>
> **2. Addressing Your Specific Questions**
>
>
> - **"Does Sequential use a Causal Mask for multiple forward passes?" Yes.** It is standard autoregressive generation (ancestral sampling), requiring $L$ forward passes with a causal mask.
>
>
> - **"Does Parallel perform a single forward pass without a mask?" No.** It performs a single forward pass, but it still uses a causal mask. The teacher must respect autoregressive constraints to calculate the correct logits for the ground-truth tokens.
> - **"Why does Sequential Extraction perform worse?"** For the same sequence x, the teacher logits are essentially identical whether computed via Sequential (many passes with causal masks) or Parallel (single batched pass with the same causal structure). The discrepancy in performance (worse PPL) arises because **Sequential extraction** requires the model to train on teacher-generated text, which tends to be **narrower and can inherit the teacher’s sampling biases**; whereas parallel extraction allows the model to train on human text with $\xi$ recovered via the posterior, with **better diversity and distributional properties**. Therefore, in Table 4 we can see that sequential extraction does not always improve over the baseline (MAUVE better, PPL worse), due to the **inherently different training corpus**.

---

### Official Review · Reviewer_NcsQ · 2025-10-28

**Soundness:** 3
**Presentation:** 4
**Contribution:** 3
**Rating:** 8
**Confidence:** 3

**Summary:**

This paper proposes a method for distilling the knowledge of an autoregressive model into a parallel/multi-token model via gumbel distillation. This key idea allows for a deterministic supervised setting for the parallel student, and effectively reframes the parallel learning task. Through various experiments, the authors demonstrate how gumbel distillation improves the generation quality of parallel decoding language models, closing the gap with the slower but more high-quality autoregressive models.

The authors validate their approach by integrating it into masked diffusion language models and multi-token prediction architectures. Quantitative results demonstrate significant and consistent gain over the original methods, on datasets such as OpenWebText and LM1B. This, along with a strong increase in the acceptance rate for the MTP setting, clearly indicates the benefits of the proposed approach.

**Strengths:**

The paper is very well written, and it is clear that the authors have put a lot of time and care into the presentation. This extends to figures, tables and the appendix as well.

The method is (to at least my understanding) sound, and the explanation is pedagogical with consistent notation. The empirical results are persuasive in that they are both consistent and strong. The tasks selected for evaluation seem relevant, and give a fairly comprehensive overview of the expected gains from the gumbel distillation approach.

**Weaknesses:**

The authors currently don’t demonstrate any results or arguments for the scalability of this approach. Considering that a major component of how well LLM works is their ability to scale, such an addition would further strengthen this paper. The only part relevant to this seems to be the brief discussion regarding the scalability of vocabulary size. Personally, I’d prefer to see an experiment where the number of model parameters are scaled in the main paper, rather than the existing ablation studies.

Less an issue and more a suggestion of structure. The current layout flow of: 2. Background, 3. Method, 4. Related Work, seems a bit unconventional? As a reader I found it a bit jarring to suddenly be exposed to a related work section after having just read the method. I would suggest considering moving the related work section before the method section or to the end of the paper.

**Questions:**

What are your intuitions regarding the scalability of this approach? In regards to model parameters, context length, dataset size etc?

---

> ### Author Response · Authors · 2025-11-23
> **Response to Question 1**
>
> We thank the reviewer for the very positive and constructive feedback, and for highlighting both the clarity and the empirical strength of the paper. We will address the remaining concerns below.
>
> **Question 1: Scalability on model size, context length, dataset size**
>
> Regarding the question on scalability with respect to model parameters, context length, and dataset size, we agree that making our intuitions and evidence more explicit will strengthen the paper.
>
> Following previous works [1,2], we already evaluate at two different **context lengths** and two different **dataset scales**: LM1B (news-based English text, ~4.4GB of text) with short sequences (e.g., 128 tokens) and OpenWebText (web-scraped content, ~38GB of text) with longer contexts (e.g., 1024 tokens). Across both, Gumbel Distillation consistently improves GenPPL and narrows the gap to the AR teacher (Table 1). Additionally, when training on OWT, the relative performance improvement of Gumbel Distillation persists across different stages of training (e.g. 10% vs. 100% tokens seen by model).
>
> Nevertheless, we conducted experiments on model parameter scaling and show partial results below. The results reported in the paper are from small-sized models with 12 Transformer blocks and embedding dimension of 768; the medium-sized models have 24 Transformer blocks and an embedding dimension of 1024. Evaluation protocols are the same, but note that our medium-sized models here are only partially trained (about 40% of tokens seen as fully trained small-sized models). And we can still observe a similar upwards trend in performance of Gumbel models over its baseline.
> | Method                           | MAUVE Score ↑ | Gen. PPL ↓ |
> |----------------------------------|---------------|------------|
> | MDLM - Small                            | 0.179         | 78.74      |
> | MDLM - Small + Gumbel Distillation       | 0.264         | 67.64      |
> | MDLM - Medium (Partially Trained)                            |  0.186        |  74.22     |
> | MDLM - Medium + Gumbel Distillation  (Partially Trained)          | 0.253      |  61.34     |
>
> **References:**
>
> [1] Subham Sahoo, Marianne Arriola, Yair Schiff, Aaron Gokaslan, Edgar Marroquin, Justin Chiu, Alexander Rush, and Volodymyr Kuleshov. Simple and effective masked diffusion language models. Advances in Neural Information Processing Systems, 2024.
> [2] Marianne Arriola, Aaron Gokaslan, Justin T Chiu, Zhihan Yang, Zhixuan Qi, Jiaqi Han, Subham Sekhar Sahoo, and Volodymyr Kuleshov. Block diffusion: Interpolating between autoregressive and diffusion language models. In International Conference on Learning Representations, 2025.

---

> ### Author Response · Authors · 2025-11-23
> **Response to Concern 1 and 2**
>
> **Response to Concern 1: Limitation on Vocabulary Size and Complexity**
>
> We agree that conditioning on a vector of size `vocab_size` raises natural questions about scalability. We already flag vocabulary size as a limitation in Section 6; here we clarify how the cost behaves and why we believe the issue is primarily about *optimization at extreme vocab sizes*, rather than an inherent barrier for large LMs.
>
> Our Gumbel injection adds a **single projection layer** that maps a Gumbel vector of dimension `V = vocab_size` to the hidden dimension `H`. This introduces a fixed parameter and compute cost of order
> $\mathcal{O}(V \times H)$, which is applied once per sequence, and does **not grow with model depth**. In contrast, the base transformer scales as $\mathcal{O}(L \times H^2)$, where \(L\) is the number of layers. As a result, the *relative* parameter overhead of the Gumbel pathway **decreases** as the transformer backbone model scales up.
>
> | Model Scale   | Layers \(L\) | Hidden \(H\) | Vocab Size | Total Params | Gumbel Param Overhead |
> |---------------|--------------|--------------|--------------|------------------------|------------------------|
> | GPT2-Small   | 12           | 768          | 50257 | 117M         | 32.9%                  |
> | GPT2-Large   | 36           | 1,280        | 50257| 774M         | 8.3%                   |
> | LLaMA-7B      | 32           | 4,096        | 32000 | 7,000M       | 2.9%                   |
> | LLaMA-13B     | 40           | 5,120        | 32000 | 13,000M      | 2.0%                   |
> | Qwen-7B | 32 | 4096 |  151851| 7,000M | 8.1% |
>
> As the table above shows, the relative overhead is relatively high only for very small models; for LLaMA-scale models, it drops to just a few percent. A similar reasoning applies to FLOPs: the extra projection of Gumbel noise is a tiny fraction of the total transformer compute once \(L\) and \(H\) are large.
> Regarding **optimization**, at the GPT-2–scale vocabularies used in our experiments we **do not observe instability or slower convergence** compared to the baseline training curves; in practice the added conditioning behaves like a small, well-behaved adapter.
>
> In the revision, we will make the above complexity analysis explicit.
>
>
> **Response to Concern 2: Paper Layout and Section Ordering**
>
>
> Regarding the suggestion about the section order, we agree that the current layout (Background - Method - Related Work) is somewhat unconventional and could benefit from re-arrangement.
>
> In the revision, we have moved Related Work either to precede the Method section (Background - Related Work - Method).

---

### Official Review · Reviewer_pK43 · 2025-10-29

**Soundness:** 3
**Presentation:** 3
**Contribution:** 2
**Rating:** 4
**Confidence:** 3

**Summary:**

This paper proposes Gumbel Distillation, a knowledge distillation framework that improves the learning of parallel decoders such as Masked Diffusion Language Models (MDLM) and Multi-Token Prediction (MTP) models. The key idea is to convert the stochastic sampling process of an autoregressive (AR) teacher into a deterministic mapping via the Gumbel-Max trick, allowing the student model to condition on the teacher’s latent Gumbel noise. This effectively transforms the hard problem of matching a complex joint token distribution into a simpler supervised learning problem. The framework is architecture-agnostic and can be plugged into various diffusion or parallel decoding models.

**Strengths:**

1. The paper correctly identifies that the main challenge of parallel decoding lies in learning the joint token dependencies. By introducing Gumbel noise as an explicit conditioning variable, the proposed approach offers a conceptually clean way to transfer dependency structure from AR to parallel models.
2. The idea of externalizing stochasticity to simplify the learning problem is an important insight likely to inspire future work in distillation and non-AR training.
3. The method is designed to integrate with different architectures (MDLM, BD3-LM, Medusa) without architectural redesign, demonstrating good generality.

**Weaknesses:**

1. Limited and marginal empirical gains. In particular, Figure 3 shows that the NFE–quality trade-off is not clearly improved—on one dataset the curve even slightly underperforms the baseline, and on the others the advantages are only marginal. The improvements on reasoning and QA benchmarks (e.g., BoolQ, ARC) are also small.
2. The paper does not compare with few-step DLM acceleration baselines such as APD (Adaptive Parallel Decoding, arXiv:2506.00413).
3. Although the approach simplifies distribution matching conceptually, the student still faces a harder supervised task: mapping high-dimensional Gumbel noise vectors to text sequences. As the vocabulary grows, the conditioning dimension scales linearly, and the paper acknowledges this as a limitation (§6). The work could benefit from more discussion or experiments on how this affects optimization stability and convergence.
4. While distillation transfers AR dependencies, it is unclear whether the Gumbel-conditioned student retains the versatility of standard masked diffusion models — e.g., controllability, flexible conditional generation. Conditioning on a fixed “blueprint” may overfit to AR-like sampling behavior and reduce adaptability.

**Questions:**

See Weakness.

---

> ### Author Response · Authors · 2025-11-23
> **Response to Concern 1**
>
> We thank the reviewer’s recognition of the conceptual significance and flexibility of our approach. Below we will address each of your concerns.
>
> **Response to Concern 1: Limited and Marginal Empirical Gains**
>
> We appreciate the reviewer's detailed examination of our results. Looking at individual numbers in isolation could make the empirical gains appear less significant. We respectfully suggest that a holistic view of the results across metrics and datasets demonstrates a substantial and consistent improvement.
>
> **Significant Improvements in Generation Quality (Table 1):** In our main results (Table 1), the relative gap-closure vs. AR is non-trivial and consistent. For example, on OpenWebText our method improves MDLM’s MAUVE by +30.0% and reduces Gen. PPL by −10.5% over the MDLM baseline, with similar relative gains on LM1B and for BD3-LM.
>
> **Improved PPL-NFE Frontier (Figure 3):** In Fig. 3, we plot Gen. PPL vs. NFE for all models under the same sampler, and the Gumbel-distilled curves lie below the corresponding baselines, showing a better quality–compute frontier.
>
> **Uniform and Consistent Improvements on Downstream Tasks:** While absolute accuracy jumps on specific QA benchmarks may seem modest, the improvements are uniformly positive across eight distinct tasks. Crucially, these gains are achieved without any task-specific fine-tuning or architectural changes—only by modifying the training objective to better align with the AR teacher.
> In summary, our method offers a simple, architecture-agnostic modification that systematically narrows the gap between parallel and AR models across all tested dimensions.
> Overall, with a simple, architecture-agnostic training modification, Gumbel Distillation systematically narrows this gap across models and datasets across all tested dimensions.

---

> ### Author Response · Authors · 2025-11-23
> **Response to Concern 2**
>
> **Response to Concern 2: Lack of Comparison Baselines for Distillation**
>
> We agree that showing how Gumbel Distillation compares with classical AR to NAR distillation will strengthen the empirical claim of our paper.
>
>  Following this suggestion, we have run new experiments to compare Gumbel Distillation against several baselines, including APD and other common distillation techniques for LLMs:
>
> - **Token-level distillation (“soft” KD)**: the student is trained to match the teacher’s full logit distribution at each position via a KL divergence term in addition to the standard loss, as in classical knowledge distillation [1].
>
> - **Sequence-level distillation (“hard” KD)**: the student is trained on sequences generated by the AR teacher (teacher continuations are treated as ground-truth labels), following sequence-level knowledge distillation [2,3].
>
> - **APD (Adaptive Parallel Decoding)**: a recent, training-free method for accelerating diffusion LMs by using a small AR verifier to adaptively accept the longest high-quality prefix [5].
>
> All methods are evaluated on MDLM trained on LM1B with the same backbone and evaluation protocol as in the main paper. The new results are as follows:
>
> | Method                           | MAUVE Score ↑ | Gen. PPL ↓ |
> |----------------------------------|---------------|------------|
> | MDLM                             | 0.179         | 78.74      |
> | MDLM + Token-level Distill       |      0.166         | 95.88      |
> | MDLM + Sequence-level Distill    | 0.169         | 99.48      |
> | MDLM + APD                    |    0.203           |  57.61        |
> | MDLM + Gumbel Distillation       | 0.264         | 67.64      |
> | MDLM + Gumbel Distillation + APD |    0.255         |   49.28       |
>
>  These results above demonstrate that:
>
> 1. **Token-level distillation** fails to address the core issue, only forcing the non-AR student to align with the AR teacher's per-token marginals, under the same conditional-independence assumption within a parallel block. In our experiments, it even **degrades** Gen. PPL relative to the baseline, showing that matching marginals alone is not sufficient for robust joint modeling and can even contradict the original learning objective.
>
> 2. **Sequence-level distillation**, where the student is trained on hard teacher-generated tokens, shows only minor to no gains in quality compared to Token-level distillation (MAUVE slightly below the baseline and much worse Gen. PPL). We hypothesize this is because teacher-generated corpora tends to be narrower, and the student can inherit teacher sampling artifacts and suffer from mode collapse.
>
>
> 3. **Gumbel Distillation**, in contrast, significantly improves both MAUVE and Gen. PPL over the original MDLM model and over both distillation baselines. Rather than distilling the outcomes (logits or generated text), it distills the **sampling process** itself: the student learns a deterministic mapping  $(x_{\neg I}, \xi_I) \mapsto x_I,$  where $\xi_I$ is the Gumbel “blueprint,” which serves as a shared latent that coordinates all tokens within a block and directly targets the joint dependencies that **classical methods do not address**.
>
> 4. **Adaptive Parallel Decoding (APD)** is ** conceptually different and complementary to our method **: it is an **inference-time** trick, not used for non-AR model distillation. It can be applied on top of both the baseline MDLM and our Gumbel-distilled MDLM without retraining the backbone. In our extended experiments, we observe that APD + Gumbel Distillation yields the best Gen PPL, as APD benefits from the stronger proposals learned by our training-time method. However, this comes at a modest cost of diversity, as evidenced by the slightly worse MAUVE score.
>
> **References:**
>
>
> [1] Hinton, Geoffrey, Oriol Vinyals, and Jeff Dean. *Distilling the knowledge in a neural network.* arXiv preprint arXiv:1503.02531 (2015).
>
> [2] Kim, Yoon, and Alexander M. Rush. *Sequence-level knowledge distillation.* Proceedings of the 2016 Conference on Empirical Methods in Natural Language Processing (EMNLP), 2016.
>
> [3] Jiatao Gu, James Bradbury, Caiming Xiong, Victor O. K. Li, and Richard Socher. *Non-autoregressive neural machine translation.* arXiv preprint arXiv:1711.02281, 2017.
>
> [4] Jungo Kasai, Jiatao Gu, Marjan Ghazvininejad, Luke Zettlemoyer, and Mohit Iyyer. *A disentangled context approach to non-autoregressive neural machine translation.* International Conference on Learning Representations (ICLR), 2021.
>
> [5] Israel, Daniel, Guy Van den Broeck, and Aditya Grover. *Accelerating Diffusion LLMs via Adaptive Parallel Decoding.* arXiv preprint arXiv:2506.00413 (2025).

---

> ### Author Response · Authors · 2025-11-23
> **Response to Concern 3**
>
> **Response to Concern 3.1: The Inherent Difficulty of Learning from Gumbel Noise**
>
>
> We respectfully posit that our method reframes the task from a harder, ill-posed problem to an easier, fully-supervised one.
> - **Original Task (Intractable)**: The baseline model must learn to model the complex, implicit joint distribution $p(x^{\mathcal{I}}|x^{-\mathcal{I}})$ over a space of $V^{|\mathcal{I}|}$ possibilities. This is the core, unsolved challenge of parallel decoding.
> - **Our Task (Tractable)**: Our model learns a fully-supervised mapping $f(x^{-\mathcal{I}}, \xi^{\mathcal{I}}) \to x^{\mathcal{I}}$. The Gumbel noise $\xi^{\mathcal{I}}$ is not just a high-dimensional input; it is a rich, deterministic "blueprint" that guides the model to the correct answer.
> If the reviewer's concern regarding difficulty were realized, we would expect poorer performance than the baselines (without using Gumbel noise). However, our empirical results show the opposite: the Gumbel-conditioned model converges successfully and achieves improvements (e.g., +30% MAUVE, +4x efficiency gain). This is strong evidence that the Gumbel signal is highly effective and makes the original problem more tractable, not less.
>
>
>
>
> **Response to Concern 3.2: Limitation on Vocabulary Size and Complexity**
>
> We agree that conditioning on a vector of size `vocab_size` raises natural questions about scalability. We already flag vocabulary size as a limitation in Section 6; here we clarify how the cost behaves and why we believe the issue is primarily about *optimization at extreme vocab sizes*, rather than an inherent barrier for large LMs.
>
> Our Gumbel injection adds a **single projection layer** that maps a Gumbel vector of dimension `V = vocab_size` to the hidden dimension `H`. This introduces a fixed parameter and compute cost of order
> $\mathcal{O}(V \times H)$, which is applied once per sequence, and does **not grow with model depth**. In contrast, the base transformer scales as $\mathcal{O}(L \times H^2)$, where `L` is the number of layers. As a result, the *relative* parameter overhead of the Gumbel pathway **decreases** as the transformer backbone model scales up.
>
> | Model Scale   | Layers \(L\) | Hidden \(H\) | Vocab Size | Total Params | Gumbel Param Overhead |
> |---------------|--------------|--------------|--------------|------------------------|------------------------|
> | GPT2-Small   | 12           | 768          | 50257 | 117M         | 32.9%                  |
> | GPT2-Large   | 36           | 1,280        | 50257| 774M         | 8.3%                   |
> | LLaMA-7B      | 32           | 4,096        | 32000 | 7,000M       | 2.9%                   |
> | LLaMA-13B     | 40           | 5,120        | 32000 | 13,000M      | 2.0%                   |
> | Qwen-7B | 32 | 4096 |  151851| 7,000M | 8.1% |
>
> As the table above shows, the relative overhead is relatively high only for very small models; for LLaMA-scale models, it drops to just a few percent. A similar reasoning applies to FLOPs: the extra projection of Gumbel noise is a tiny fraction of the total transformer compute once `L` and `H` are large.
> Regarding **optimization**, at the GPT-2–scale vocabularies used in our experiments we **do not observe instability or slower convergence** compared to the baseline training curves; in practice the added conditioning behaves like a small, well-behaved adapter.
>
> In the revision, we will make the above complexity analysis explicit.

---

> ### Author Response · Authors · 2025-11-23
> **Response to Concern 4**
>
> **Response to Concern 4: Possible overfitting to AR behavior / loss of versatility**
>
> We thank the reviewer for raising this important point. We agree that retaining the flexible controllability of standard MDLMs is crucial. We are happy to clarify that Gumbel Distillation is designed to improve coherence without sacrificing the versatility of the masked diffusion framework.
>
> First, **Gumbel Distillation does not change the decoding strategy** of the student. The model remains a parallel/masked decoder: it predicts a block of tokens given masked context, and at inference we can freely choose the masking schedule and conditional setup (e.g., infilling, prefix-only generation) as in standard MDLM/BD3-LM. The Gumbel noise \(\xi\) is a **latent variable** that acts as an extra input that helps model joint structure inside a block, and there is no constraint that forces the student to follow the teacher’s AR sampling procedure at test time.
>
> To directly test whether the Gumbel-conditioned model loses flexibility on a **non-AR task**, we ran an **infilling experiment** on the LM1B validation set, where 1) We first take sequences of length 128 from LM1B validation set; 2) For each sequence, we randomly mask out 32 tokens uniformly at random over positions; 3)  The model is given the partially observed sequence and asked to infill the masked tokens using the **same MDLM sampler** as in the main experiments; 4) We finally compute Gen PPL only on the generated (masked) positions, using the same AR evaluator as in Table 1. Averaged over 300 samples, we have:
>
> | Model                      | Gen. PPL on infilled tokens ↓ |
> |----------------------------|-------------------------------|
> | MDLM (baseline)            | 55.1                          |
> | MDLM + Gumbel Distillation | **47.8**                      |
>
> Despite being evaluated in a **pure infilling setting** (no AR teacher in the loop at test time, non-sequential masking), the Gumbel-distilled model achieves lower Gen. PPL on the masked spans than the baseline MDLM, in a similar trend as pure unconditional generation. This indicates that the blueprint does not harm performance on a standard masked-LM style task, which is exactly where a loss of versatility would be expected to show up if the model had overfit to AR behavior.
>
> To summarize, we would like to point out that: Gumbel Distillation leaves the **decoding interface** of MDLM/BD3-LM unchanged. The Gumbel blueprint is a **latent, tunable signal** rather than a hard AR template, and empirically, the Gumbel-conditioned student continues to perform well on non-AR tasks such as masked infilling, rather than overfitting to a particular AR sampling pattern.

---

### Official Review · Reviewer_7yqW · 2025-11-01

**Soundness:** 3
**Presentation:** 3
**Contribution:** 2
**Rating:** 4
**Confidence:** 4

**Summary:**

This paper addresses a critical trade-off in language model decoding: autoregressive (AR) models achieve high generation quality by modeling sequential token dependencies via chain-rule factorization but suffer from slow, token-by-token inference; parallel (non-autoregressive) decoders (e.g., MDLM, BD3-LM, Medusa) accelerate inference by generating multiple tokens simultaneously but struggle to capture the complex joint distribution of token sequences, leading to degraded coherence, repetition, or grammatical errors.

To bridge this gap, the authors propose Gumbel Distillation, a model-agnostic knowledge distillation framework that enables parallel decoders to learn the joint distribution from a high-performance AR "teacher" model. The core insight leverages the Gumbel-Max trick—a re-parameterization tool for categorical sampling—to establish a deterministic mapping between latent Gumbel noise vectors and the teacher’s output token sequences.

**Strengths:**

1. The paper directly addresses the long-standing trade-off in language model decoding: parallel decoders sacrifice generation quality (due to poor joint token distribution modeling) for speed, while autoregressive (AR) models excel at quality but are slow.
2. The introduced knowledge distillation mechanism to transfer the AR teacher’s sequential dependency knowledge to parallel students. This focus on "fixing the joint distribution defect" aligns with the most critical unmet need in parallel decoding research, making the work problem-driven and practically relevant.
3. The paper’s experimental design provides controllable evidence for the effectiveness of Gumbel Distillation.

**Weaknesses:**

1. A notable omission is the lack of targeted discussion on the classical "AR-supervised NAR distillation" baseline, a well-established method in non-autoregressive decoding where NAR models are trained to mimic the sampled outputs of AR teachers via cross-entropy (CE) loss or sequence-level losses. This gap may lead readers to question whether the authors have overlooked a foundational approach in the field.
2. While the Gumbel-Max trick and knowledge distillation are individually well-established in NLP and CV, the paper does not clearly articulate why their integration for transferring AR teachers’ joint token distributions to parallel decoders is non-trivial.
3. The paper’s framing of results overpromises, while failing to contextualize the persistent quality gap between AR teachers and Gumbel-enhanced parallel models. The paper uses language like "bridge this gap" and "resolve the quality-efficiency trade-off," but the remaining performance difference between AR (e.g., GPT-2-Large) and optimized parallel models (e.g., Gumbel-MDLM) is still large.

**Questions:**

See weaknesses.

---

> ### Author Response · Authors · 2025-11-23
> **Response to Concern 1**
>
> We thank Reviewer 7yqW for their feedback and constructive suggestions. We will address each of the concern and question below:
>
> **Response to Concern 1 (AR-supervised NAR distillation baselines):**
>
>
> We agree that our paper could benefit from direct comparison with classical AR-supervised NAR distillation baselines as well as discussion on their differences. Following this suggestion, we have run new experiments to compare Gumbel Distillation against:
>
> - **Token-level distillation (“soft” KD)**: the student is trained to match the teacher’s full logit distribution at each position via a KL divergence term in addition to the standard loss, as in classical knowledge distillation [1].
>
> - **Sequence-level distillation (“hard” KD)**: the student is trained on sequences generated by the AR teacher (teacher’s generations are treated as ground-truth labels), following sequence-level knowledge distillation [2,3].
>
> - **APD (Adaptive Parallel Decoding)** (as suggested by Reviewer pK43), a recent, training-free method for accelerating diffusion LMs by using an extra, small AR verifier to adaptively accept the longest high-quality generated sequence[4].
>
> All methods are evaluated on MDLM trained on LM1B with the same backbone and evaluation protocol as in the main paper. The new results are as follows:
>
> | Method                           | MAUVE Score ↑ | Gen. PPL ↓ |
> |----------------------------------|---------------|------------|
> | MDLM                             | 0.179         | 78.74      |
> | MDLM + Token-level Distill       |      0.166         | 95.88      |
> | MDLM + Sequence-level Distill    | 0.169         | 99.48      |
> | MDLM + APD [5]                   |    0.203           |  57.61        |
> | MDLM + Gumbel Distillation       | 0.264         | 67.64      |
> | MDLM + Gumbel Distillation + APD |    0.255         |   49.28       |
>
> As the new results demonstrate:
>
> 1. **Token-level distillation** fails to address the core issue, only forcing the non-AR student to align with the AR teacher's per-token marginals, under the same conditional-independence assumption within a parallel block. In our experiments, it even **degrades** Gen. PPL relative to the baseline, showing that matching marginals alone is not sufficient for robust joint modeling and can even contradict the original learning objective.
>
> 2. **Sequence-level distillation**, where the student is trained on hard teacher-generated tokens, shows only minor to no gains in quality compared to Token-level distillation (MAUVE slightly below the baseline and much worse Gen. PPL). We hypothesize this is because teacher-generated corpora tends to be narrower, and the student can inherit teacher sampling artifacts and suffer from mode collapse.
>
>
> 3. **Gumbel Distillation**, in contrast, significantly improves both MAUVE and Gen. PPL over the original MDLM model and over both distillation baselines. Rather than distilling the outcomes (logits or generated text), it distills the **sampling process** itself: the student learns a deterministic mapping  $(x_{\neg I}, \xi_I) \mapsto x_I,$  where $\xi_I$ is the Gumbel “blueprint,” which serves as a shared latent that coordinates all tokens within a block and directly targets the joint dependencies that **classical methods do not address**.
>
> 4. **Adaptive Parallel Decoding (APD)** can be applied on top of both the baseline MDLM and our Gumbel-distilled MDLM without retraining the backbone. In our extended experiments, we observe that APD + Gumbel Distillation yields the best Gen PPL, as APD benefits from the stronger proposals learned by our training-time method. However, this comes at a modest cost of diversity, as evidenced by the slightly worse MAUVE score.
>
>
> **References:**
>
> [1] Hinton, Geoffrey, Oriol Vinyals, and Jeff Dean. *Distilling the knowledge in a neural network.* arXiv preprint arXiv:1503.02531 (2015).
>
> [2] Kim, Yoon, and Alexander M. Rush. *Sequence-level knowledge distillation.* Proceedings of the 2016 Conference on Empirical Methods in Natural Language Processing (EMNLP), 2016.
>
> [3] Jiatao Gu, James Bradbury, Caiming Xiong, Victor O. K. Li, and Richard Socher. *Non-autoregressive neural machine translation.* arXiv preprint arXiv:1711.02281, 2017.
>
> [4] Jungo Kasai, Jiatao Gu, Marjan Ghazvininejad, Luke Zettlemoyer, and Mohit Iyyer. *A disentangled context approach to non-autoregressive neural machine translation.* International Conference on Learning Representations (ICLR), 2021.
>
> [5] Israel, Daniel, Guy Van den Broeck, and Aditya Grover. *Accelerating Diffusion LLMs via Adaptive Parallel Decoding.* arXiv preprint arXiv:2506.00413 (2025).

---

> ### Author Response · Authors · 2025-11-23
> **Response to Concern 2 and 3**
>
> **Response to Concern 2 (Novelty of Integrating Gumbel-Max and Distillation):**
>
> We thank the reviewer for this opportunity to clarify our contribution. While Gumbel-Max and Knowledge Distillation (KD) are indeed established techniques, their integration in our framework addresses a specific, non-trivial challenge: **transferring the complex joint dependencies of an AR teacher to a parallel decoder without autoregression.**
> We clarify the distinct novelty of our integration on two levels:
> 1. **Re-purposing Gumbel-Max (From Sampling to Deterministic Mapping):** Standard applications of the Gumbel-Max trick in NLP/CV typically use it for forward sampling (i.e., reparameterization for categorical sampling). **The Non-Trivial Insight** is that our method **repurposes this trick to** create a **deterministic mapping** from latent Gumbel noise $\xi^{\mathcal{I}}$ to the teacher’s observed token $x^{\mathcal{I}}$. This allows the student model to learn from the mapping rather than just the output. To our knowledge, using the Gumbel trick to recover and condition on latent noise for text generation is unexplored in prior work.
> 2. **Distillation via Latent Conditioning**:  Standard AR-to-NAR distillation typically matches outcomes (either per-token marginals or generated sequences). In comparison, we move beyond distilling the outcome (logits/text). Instead, **we condition the parallel student on the teacher’s recovered latent code (Gumbel noise).** This effectively captures the AR teacher's structure in a parallel format. This capability has not been present in standard outcome-based KD methods.
>
> **Response to Concern 3 (phrasing/overpromising vs remaining AR–parallel gap):**
>
> We appreciate this observation and agree that some of our phrasing in the current draft may be too strong given the remaining gap to the AR teacher. Our intended claim is **not** to say that Gumbel Distillation completely closes the gap or fully “resolves” the quality–efficiency trade-off, but rather that it systematically narrows the AR–parallel gap across architectures and datasets. In the revision, we have toned down the language in our paper, and replace phrases such as “bridge this gap” and “resolve the quality–efficiency trade-off” with more accurate statements like **“narrow the gap between AR and parallel decoders”**. We hope these changes will make our positioning more precise.

---

### Author Response · Authors · 2025-11-26
**Follow-up on Rebuttals**

Dear Reviewers,

We would like to thank you again for the time and care you put into reviewing our submission. Following your comments, we have added new experiments (e.g., classical KD and APD baselines, larger-scale MTP results, infilling evaluations, and vocabulary/complexity analysis) and clarified several methodological points in our rebuttal and revision plan.

If you have a moment, we would be very grateful if you could take a look and let us know whether our responses address your main concerns, and whether there are any remaining issues we should clarify. If you feel that the additional results and clarifications meaningfully change your overall assessment, we would appreciate it very much if you could update your review accordingly.

Thank you again for your thoughtful feedback and for helping us improve the paper.

Authors

---

### Comment · Area_Chair_rfJY · 2025-11-28
**Please help check the rebuttal**

Dear Reviewers,

Thanks for your effort in reviewing the manuscript. Now the authors have provided the rebuttal and it's highly recommended to take a look and give your feedback. Thanks.

AC.

---

### Meta-Review · Area_Chair_fr5u · 2026-01-07

**Summary:**

This paper proposes Gumbel Distillation, a knowledge distillation framework that enables parallel decoders to learn joint token distributions from autoregressive teachers by leveraging the Gumbel-Max trick to create deterministic mappings from latent noise to output tokens. The method is model-agnostic and integrates with diverse architectures (MDLM, BD3-LM, Medusa). Experiments on LM1B and OpenWebText show substantial improvements in generation quality.

Initial scores were 4/4/8/4. Authors provided comprehensive responses addressing major concerns with new experiments. No reviewers engaged during discussion.

**Reviewer Concerns:**

Concerns successfully addressed:
- Lack of Baselines (7yqW, pK43, B2R6).
Authors conducted new experiments comparing Gumbel Distillation against Token-level KD and Sequence-level KD (suggested by Reviewer 7yqW), as well as Adaptive Parallel Decoding (APD) (suggested by Reviewer pK43) on the MDLM backbone. Gumbel Distillation significantly outperforms all baselines and the new results also demonstrated that combining Gumbel Distillation with APD yields the best overall performance.
- Scalability & Vocabulary Size (pK43, NcsQ, B2R6).
Authors provided detailed complexity analysis showing Gumbel projection overhead decreases with model scale. Also, authors applied their method to Vicuna-7B (using the Medusa architecture) and observed consistent improvements in acceptance rates (e.g., +28.1% relative gain on Head 2), proving the method scales effectively to 7B+ parameter models.
- Overfitting to AR behavior and limited empirical gains (pK43).
Authors provided infilling experiment showing Gumbel-distilled model maintains flexibility on non-AR tasks, and also clarified that while individual numbers may appear modest, gains are substantial in relative terms and consistent across all metrics and datasets.
- Novelty of combining Gumbel-Max and KD (7yqW).
Authors clarified non-trivial contributions: Repurposing Gumbel-Max from forward sampling to deterministic mapping recovery and distilling sampling process via latent conditioning rather than outcomes.

Outstanding concerns:
- None. I think the rebuttal well addressed almost all major concerns.

**Reviewer Scores:**

- Reviewer 7yqW: Increases from 4 to 6.
- Reviewer pK43: Increases from 4 to 6.
- Reviewer NcsQ: Remains at 8.
- Reviewer B2R6: Increases from 4 to 6.

---

### Decision · Program_Chairs · 2026-01-26

Accept (Poster)